# Alkyladenine DNA glycosylase associates with transcription elongation to coordinate DNA repair with gene expression

Nicola P. Montaldo[1,2], Diana L. Bordin[1,9], Alessandro Brambilla[1,9], Marcel Rösinger[2], Sarah L. Fordyce Martin[1], Karine Øian Bjørås[1], Stefano Bradamante[1], Per Arne Aas[1], Antonia Furrer[2,8], Lene C. Olsen [1,3,4], Nicolas Kunath[1], Marit Otterlei[1], Pål Sætrom [1,3,4,5], Magnar Bjørås[1,6], Leona D. Samson[7] & Barbara van Loon [1,2]*

Base excision repair (BER) initiated by alkyladenine DNA glycosylase (AAG) is essential for removal of aberrantly methylated DNA bases. Genome instability and accumulation of aberrant bases accompany multiple diseases, including cancer and neurological disorders. While BER is well studied on naked DNA, it remains unclear how BER efficiently operates on chromatin. Here, we show that AAG binds to chromatin and forms complex with RNA polymerase (pol) II. This occurs through direct interaction with Elongator and results in transcriptional co-regulation. Importantly, at co-regulated genes, aberrantly methylated bases accumulate towards the 3'end in regions enriched for BER enzymes AAG and APE1, Elongator and active RNA pol II. Active transcription and functional Elongator are further crucial to ensure efficient BER, by promoting AAG and APE1 chromatin recruitment. Our findings provide insights into genome stability maintenance in actively transcribing chromatin and reveal roles of aberrantly methylated bases in regulation of gene expression.

[1] Department of Clinical and Molecular Medicine, Faculty of Medicine and Health Sciences, Norwegian University of Science and Technology (NTNU), 7491 Trondheim, Norway. [2] Department of Molecular Mechanisms of Disease, University of Zurich, 8057 Zurich, Switzerland. [3] Bioinformatics core facility – BioCore, Norwegian University of Science and Technology (NTNU), 7491 Trondheim, Norway. [4] K.G. Jebsen Center for Genetic Epidemiology, Norwegian University of Science and Technology (NTNU), 7491 Trondheim, Norway. [5] Department of Computer Science, Faculty of Information Technology and Electrical Engineering, Norwegian University of Science and Technology (NTNU), 7491 Trondheim, Norway. [6] Department of Microbiology, Oslo University Hospital, University of Oslo, 0372 Oslo, Norway. [7] Department of Biological Engineering, Department of Biology, David H. Koch Institute of integrative Cancer Research, Massachusetts Institute of Technology, Cambridge, MA 02139, USA. [8] Present address: Paul Scherrer Institute, 5232 Villigen, Switzerland. [9] These authors contributed equally: Diana L. Bordin, Alessandro Brambilla. *email: barbara.v.loon@ntnu.no

Thousands of DNA base lesions are generated in each cell of our body every day as a consequence of exposure to endogenous and exogenous DNA damaging agents[1]. Aberrantly methylated bases account for a large proportion of those lesions. Estimated steady-state levels of the most frequent aberrantly methylated bases: 7-methylguanine (7meG) and 3-methyladenine (3meA), are 6,000 and 1,200 lesions per mammalian cell per day respectively[2]. Several lines of evidence strongly indicate that accumulation of aberrant DNA bases and genome instability accompany major human diseases like cancer, inflammatory diseases and neurological conditions[3,4].

Base excision repair (BER) is a highly efficient mechanism for the removal of aberrant DNA bases. This multi-step repair process is initiated by substrate-specific DNA glycosylases that recognize and remove aberrant bases[3,5]. The resulting abasic (AP) site is processed by the apurinic/apyrimidinic endonuclease 1 (APE1) that generates a single-strand break (SSB), which is subsequently filled by DNA polymerase (pol) β, and the resulting nick sealed by DNA ligase III/XRCC1 (ref. [6]). The BER specificity is determined by the type of DNA glycosylase that initiates the pathway[7]. Alkyladenine DNA glycosylase (AAG; also known as N-methylpurine-DNA glycosylase, MPG) is the major mammalian DNA glycosylase responsible for the removal of aberrantly methylated bases 7meG and 3meA[6,7]. Besides abundant 3meA and 7meG, AAG is able to act on several other aberrant bases including hypoxanthine (Hx) and 1,N6-ethenoadenine[3]. AAG participates in both nuclear and mitochondrial DNA repair[8]. Much of our current knowledge about the different BER steps derives from in vitro studies using naked DNA as a substrate. However, eukaryotic cells need to repair DNA in the context of chromatin. Recent work demonstrated that the activity of human AAG, and of several other DNA glycosylases, is strongly impaired in the context of nucleosomes[9]. AAG activity is dramatically inhibited (84–100%) when the aberrant base is positioned midway or directly faces the surface of the histone octamer. Furthermore, the presence of nucleosomes was shown to significantly impair the activity of the downstream BER proteins APE1, DNA pol β and DNA ligase III/XRCC1 (refs[10–12]). It was proposed that BER could overcome the inhibitory effect of nucleosomes by associating with processes that involve chromatin reorganization, such as replication and transcription[13–15]. This idea is further supported by the notion that BER preferentially occurs on the transcribed strand[16–18]. However, firm evidence demonstrating that BER associates with the transcription is still lacking. AAG could potentially be associated with the transcription through interaction with different transcription factors. It was shown that AAG forms a complex with a transcriptional repressor methylated DNA-binding domain 1 (MBD1) and that this complex could inhibit the transcription[19]. Furthermore, AAG was demonstrated to bind to the estrogen receptor (ER) α, which in turn stimulates AAG activity[20] and could potentially result in repression of the ERα-responsive genes. Interestingly, recent work in mice focusing on DNA glycosylases responsible for the repair of oxidized bases, similarly, suggested the role of glycosylases in regulation of ERα-gene expression[21]. Taken together, these findings indicate that coupling of AAG-initiated BER to transcription could enable more efficient repair and that AAG potentially influences gene expression.

Recent extensive protein interaction study[22] predicted that AAG forms a complex with the active transcription machinery through a possible interaction with Elongator complex. Elongator is a highly conserved complex that participates in several pathways, including facilitation of transcription elongation by interacting with hyperphosphorylated RNA pol II[23,24]. It is composed of six subunits ELP1 to ELP6 organized in a dodecamer[25,26]. Every subunit is required for the complex to function. ELP1 (also known

as IKBKAP) is the largest subunit of the complex and provides a scaffolding function[27]. ELP3 has histone acetyltransferase (HAT) activity and acetylates histone H3 (refs[28,29]). Mutations in human *ELP1*, and consequently impaired Elongator function cause neurodevelopmental disorder, and lead to reduced expression of Elongator-dependent genes in patient-derived cells[30]. Interestingly, loss of Elp1 in yeast causes hypersensitivity to methyl methanesulfonate (MMS) and hydroxyurea[31]. While the predicted association between AAG and Elongator complex suggests that AAG-initiated BER may be coordinated with transcription elongation, the interaction remains yet to be confirmed and its relevance explored. Taken together, several studies suggest that AAG could form a complex with different transcriptional components, however whether and where these interactions take place in the context of chromatin, as well as the extent to which they facilitate AAG-initiated BER and regulate gene expression remains unknown.

In this study we show that the majority of cellular AAG localizes at chromatin and is engaged in a complex with actively transcribing RNA pol II. This occurs primarily through direct interaction of the unstructured N-terminal AAG region with the ELP1 subunit of transcriptional Elongator complex. To our knowledge this is the first evidence of BER association with active transcription. RNA-sequencing experiments further show that AAG and ELP1 co-regulate genes, which are primarily repressed by AAG and stimulated by ELP1. Notably, at co-regulated genes, the endogenous aberrantly methylated DNA bases accumulate towards the 3′end in regions co-occupied by ELP1, elongating RNA pol II, and BER enzymes AAG and APE1. Chromatin recruitment of the BER enzymes is strongly dependent on Elongator presence and active transcription, since ELP1 loss as well as transcription inhibition cause globally reduced AAG and APE1 binding. Importantly, ability to interact with ELP1, presence of functional Elongator and active transcription are needed to ensure efficient AAG-initiated BER, and their inactivation results in impaired repair and significant accumulation of AAG substrates in the genome. Based on our findings, we propose that AAG, in concert with Elongator complex and active transcription, coordinates repair of aberrantly methylated DNA bases with regulation of gene expression.

## Results

**AAG associates with transcription to regulate expression.** AAG predominantly localizes to the nucleus where it recognizes and removes aberrant DNA bases. However, it remains unknown how AAG initiates repair in the context of chromatin, and in which specific regions of the genome the repair takes place. To determine which portion of nuclear AAG is bound to the chromatin and could actively participate in the repair of genomic DNA, HEK293T cells were fractionated. The amount of AAG detected in the chromatin fraction (CF) was comparable to the total AAG fraction (TF), while only low levels of AAG were detected in the soluble fraction (SF) (Fig. 1a, b), thus indicating that the majority of cellular AAG is chromatin bound. In order to efficiently repair aberrant bases in chromatin context, DNA glycosylases were suggested to associate with processes that involve chromatin reorganization, such as transcription[13]. To determine if AAG engages with the active transcription, immunoprecipitation (IP) was performed and AAG-bound cellular complexes analyzed. Interestingly, the IP analysis indicated that AAG forms a complex with active RNA pol II phosphorylated at Serine 2 of the C-terminal domain (RNA pol II S2P) (Fig. 1c). Presence of different nucleases (DNaseI, MNase and RNaseI) did not majorly affect the level of IP-ed RNA pol II S2P (Fig. 1c, compare lanes 6, 7 to 5), thus suggesting that complex formation between AAG and RNA

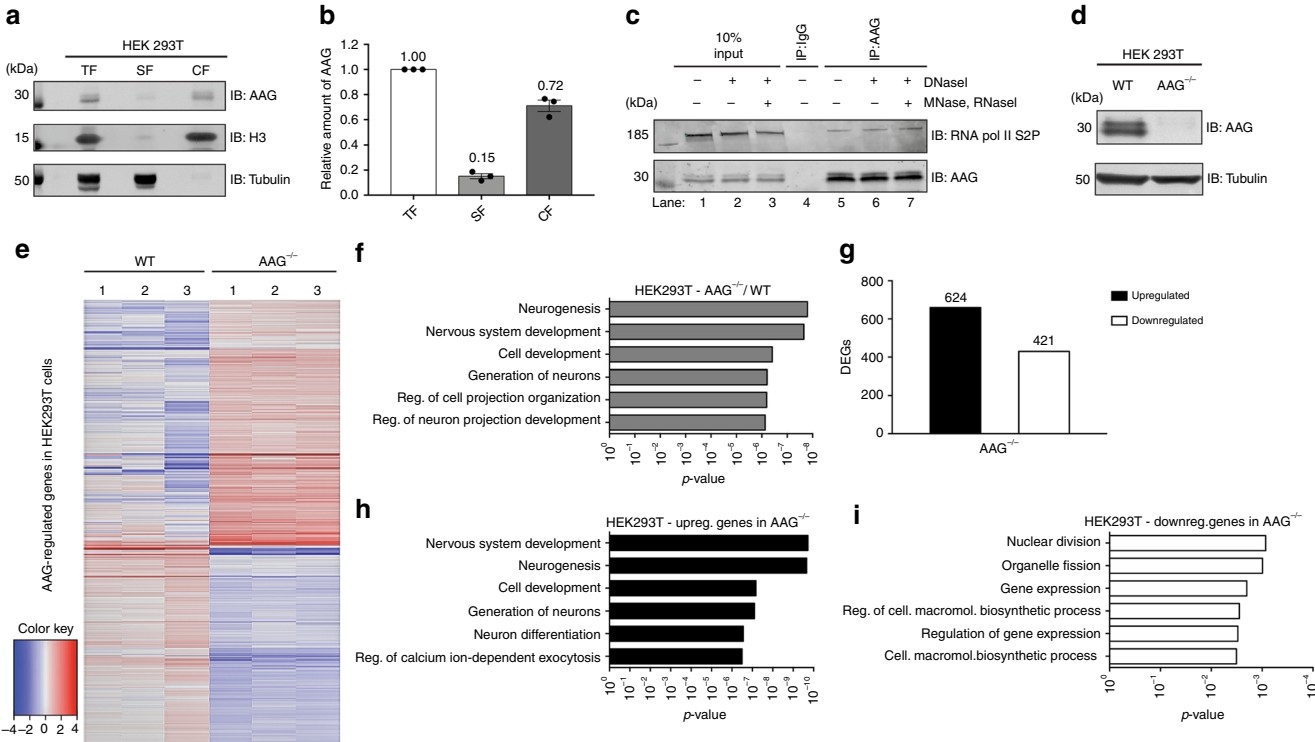

**Fig. 1** AAG associates with active transcription and regulates gene expression. **a** Immunoblot analysis of chromatin fractionation assay indicating AAG distribution in total fraction (TF), soluble supernatant fraction (SF) and chromatin fraction (CF). Histone H3 and α-tubulin served as controls. **b** Quantification of three independent experiments as the one depicted in **a**; error bars represent mean ± SEM ($n = 3$). **c** Immunoprecipitation of AAG from HEK293T whole-cell extracts (WCEs) untreated or treated with DNaseI, Mnase, and RNaseI, showing the interaction with RNA polymerase II phosphorylated at Serine 2 (S2P) of CTD repeat. **d** Immunoblot analysis of WCEs from HEK293T WT and AAG$^{-/-}$ cells generated by CRISPR-Cas9 technology. **e** Heat map of the expression levels of AAG-regulated genes in HEK293T cells. Color scale is representing change in the gene expression depicting log2 fold change relative to the mean. **f** Top six biological processes (BP) gene ontology (GO) terms as determined by the Database for Annotation, Visualization and Integrated Discovery (DAVID) for genes dysregulated in HEK293T AAG$^{-/-}$ cells when compared to WT. **g** Up- and down-regulated differentially expressed genes (DEGs) in HEK293T AAG$^{-/-}$ cells. **h**, **i** Top six BP GO terms as determined by DAVID for genes upregulated (**h**) and downregulated (**i**) genes. Source data are provided as Source Data file.

pol II is DNA and RNA independent, and likely mediated through protein-protein interactions. To determine regions in which transcription is influenced by AAG presence, we generated HEK293T cells lacking AAG (AAG$^{-/-}$) using the CRISPR-Cas9 engineering approach (Fig. 1d), and subjected them to RNA sequencing (Fig. 1e). Comparison of wild type (WT) and AAG$^{-/-}$ transcriptomes revealed that, at ≥1.5-fold change and an FDR ≤ 0.1, 1,045 genes were differentially expressed in AAG$^{-/-}$ cells. The subsequent DAVID gene ontology (GO) enrichment analysis[32] revealed that the majority of differentially expressed genes (DEGs) in AAG$^{-/-}$ belonged to neurogenesis and nervous system development processes, 12.3 and 16.8% respectively (Fig. 1f, Supplementary Data 1). These findings are further supported by analysis of DEGs in HAP1 AAG$^{-/-}$ cells, which determined the neurodevelopmental processes as the most significantly enriched (Supplementary Fig. 1, Supplementary Data 1). Subsequent separation of DEGs based on the expression change, revealed that the large portion of genes affected by the loss of AAG are upregulated (Fig. 1g). These upregulated genes were the main drivers of the GO, segregating in the nervous system development and neurogenesis processes (Fig. 1h, Supplementary Data 1), while the downregulated genes produced no significant GO terms (Fig. 1i, Supplementary Data 1). Taken together these findings suggest that AAG binds to chromatin, forms a complex with the active transcription and regulates gene expression.

**AAG directly binds the transcriptional Elongator complex.** AAG-initiated BER can be linked to transcription through direct association with RNA pol II or by binding other components of the transcription machinery. To identify proteins that form a complex with AAG, we expressed and affinity-purified the FLAG-tagged AAG from HEK293T cells, either untreated or exposed to the alkylating agent MMS. A control mock purification was performed in parallel from cells transfected with empty FLAG vector. Proteins in all samples were visualized by silver staining, indicating AAG presence in appropriate fractions (Fig. 2a). The affinity-purified samples were next subjected to liquid chromatography-tandem mass spectrometry (LC/MS-MS) analysis (Supplementary Data 2). This approach indicated ELP1 as the most enriched AAG interacting partner. In addition, ELP2 and 3 were also detected in AAG samples, thus suggesting that whole holo-Elongator complex is present. MMS exposure did not appear to affect AAG-Elongator complex formation, since a comparable number of peptide spectra were detected in the untreated and treated samples (Supplementary Data 2). Subsequent IP experiments showed that, similar to the exogenous FLAG-AAG, the endogenous AAG forms a complex with ELP1 independently of the MMS treatment (Fig. 2b). Further, the separation of protein complexes present in the HeLa whole cell extract (WCE) revealed that the main Elongator subunits co-elute with AAG (Supplementary Fig. 2a). Since both AAG and Elongator

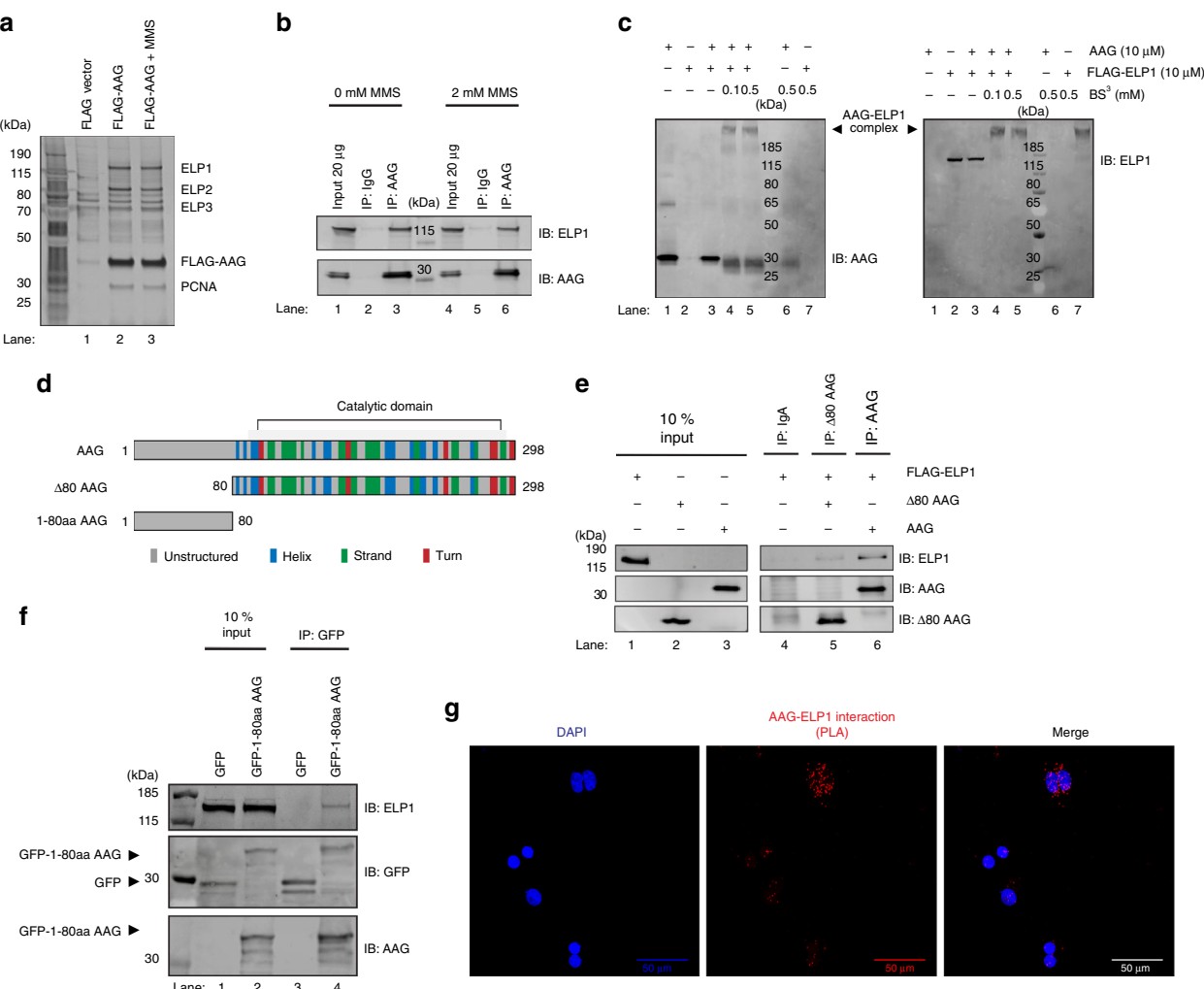

**Fig. 2** AAG directly interacts with ELP1 subunit of transcriptional Elongator complex. **a** Silver staining analysis of untreated or MMS (methyl methanesulfonate) treated FLAG-AAG immunoprecipitation (IP) samples subjected to LC/MS-MS analysis. IP sample from HEK293T cells transfected with empty FLAG vector served as negative control. LC/MS-MS analysis of FLAG-AAG IP samples identified ELP1 as the most specifically enriched AAG interacting partner. Remaining subunits of core Elongator (ELP2 and 3), and known interacting partner PCNA are depicted. See also Supplementary Data 2 for the complete list of proteins identified in all samples. **b** IP of AAG from HEK293T whole cell extracts (WCEs) treated or untreated with MMS. **c** Immunoblot analysis of gel shift experiments with purified recombinant AAG and FLAG-ELP1 proteins, and chemical crosslinking with indicated amounts of BS$^3$ (bissulfosuccinimidyl suberate). Control samples – proteins in the absence of crosslinking agent (lanes 1-3); and single proteins with highest amount of BS$^3$ (lanes 6, 7). **d** Schematic representation of full-length AAG, AAG lacking 80 N-terminal amino acids (Δ80 AAG), and first 80 N-terminal amino acids of AAG (1-80aa AAG); numbers indicate amino acids. **e** Co-IP of purified FLAG-ELP1 with AAG full length, or AAG lacking 80 N-terminal amino acids (Δ80 AAG). **f** IP of GFP-tagged first 80 N-terminal amino acids of AAG (1-80aa AAG) (GFP-1-80aa AAG) expressed in HEK293T AAG$^{-/-}$ cells. GFP-tag positioned N-terminally. **g** Proximity ligation assay (PLA) showing the AAG-ELP1 interaction in HEK293T cells. Scale bar: 50 μm. Source data are provided as a Source Data file.

are DNA-binding proteins, we next examined whether the complex formation is dependent on the presence of nucleic acids. Addition of DNaseI, MNase or RNaseI during either AAG or ELP1 IP did not influence the levels of co-precipitated proteins (Supplementary Fig. 2b, c). These findings thus suggest that in human cells endogenous AAG forms a complex with Elongator, independently of the presence of nucleic acids and exposure to genotoxic stress. To examine if AAG binds directly to Elongator, gel shift experiments were performed using purified recombinant AAG and the FLAG-tagged ELP1 (Supplementary Fig. 2d, e), and the chemical crosslinking with bissulfosuccinimidyl suberate (BS$^3$). ELP1 was targeted, since it was the predominant Elongator subunit detected by LC/MS-MS in the complex with AAG (Supplementary Data 2). Interestingly, the gel shift experiments combined with chemical

crosslinking revealed that AAG efficiently forms a complex with the ELP1 dimer, indicating direct binding of these proteins (Fig. 2c, lanes 4, 5) and further supporting earlier reports of ELP1 dimerization[33,34] (Fig. 2c, lane 7). No complex formation was observed in control reactions with crosslinker and AAG alone (Fig. 2c, lane 6). Since AAG is composed of an unstructured N-terminal region and a catalytic DNA glycosylase domain (Fig. 2d), we next addressed which AAG region is important for the interaction with ELP1. Incubation of recombinant FLAG-ELP1 with full-length AAG or AAG lacking 80 N-terminal amino acids (Δ80 AAG) (Supplementary Fig. 2f) and subsequent IP analysis, indicated that both AAG forms interact with FLAG–ELP1, although the amount of precipitated Elongator was markedly reduced with Δ80 AAG (Fig. 2e). This result suggested that the unstructured N-terminal region might

be important for AAG-ELP1 interaction. To further test this HEK293T AAG$^{-/-}$ cells were complemented with the 80 N-terminal amino acids (1-80aa) of AAG fused to a GFP tag, positioned either N- or C-terminally (GFP-1-80aa AAG and 1-80aa AAG-GFP, respectively). Subsequent IP indicated that 1-80aa AAG is sufficient to bring down ELP1 (Fig. 2f and Supplementary Fig. 2g), indicating that the unstructured N-terminal AAG region successfully binds ELP1. Next, to visualize the AAG-ELP1 interaction within cells proximity ligation assay (PLA) was performed, showing that the AAG-ELP1 interaction mainly occurs in the nucleus (Fig. 2g and Supplementary Fig. 2h, i). Interestingly, a small fraction of the signal was detected surrounding the nucleus. Taken together, these findings indicate that in the nucleus AAG binds directly to the ELP1 subunit of the transcriptional Elongator complex, predominantly through the unstructured N-terminal domain of AAG.

**AAG and Elongator co-regulate gene expression.** To identify genes and regions at which AAG and Elongator have the most significant impact we generated, in addition to AAG$^{-/-}$ cells, the HEK293T cells lacking ELP1 (ELP1$^{-/-}$), and both AAG and ELP1 (AAG$^{-/-}$ELP1$^{-/-}$) (Fig. 3a), via the CRISPR-Cas9 approach. Subsequent RNA sequencing analysis of ELP1$^{-/-}$ cells showed that loss of ELP1 results in ≥2-fold altered expression of 489 genes, with FDR ≤ 0.1. Similar to AAG$^{-/-}$ cells, 21.2% of all DEGs in ELP1$^{-/-}$ cells belonged to the nervous system

development processes (Fig. 3b, Supplementary Data 1). The majority of the DEGs were downregulated in ELP1$^{-/-}$ cells (343 out of 489) (Fig. 3c), thus supporting previous observation that Elongator promotes transcription[30]. Comparison of genes differentially expressed in AAG$^{-/-}$ and ELP1$^{-/-}$ cells resulted in identification of 113 co-regulated genes (Fig. 3d) that predominantly clustered into processes involving generation of neurons and neurogenesis (Fig. 3e, Supplementary Data 1). Interestingly, the majority of co-regulated genes were upregulated in AAG$^{-/-}$ and downregulated in ELP1$^{-/-}$ cells (Fig. 3f), thus following the general direction of all DEGs in these two cell lines (Figs. 1g and 3c). The RNA sequencing results were confirmed by qPCR measurements of mRNA levels for multiple co-regulated genes belonging to the top identified biological processes presented in Fig. 3e. The mRNA levels of ALDH1A2, CRMP1, CDH23, SYT9, CDH4, NPTX2, NOVA2, and CDH22 were, as expected, significantly increased in AAG$^{-/-}$, and decreased in ELP1$^{-/-}$ cells (Fig. 3g–i and Supplementary Fig. 3), while the level of non-affected control YTHDC1 was unchanged in all tested cell lines (Fig. 3j). The observed increase in the expression of co-regulated genes was similar in two independent AAG$^{-/-}$ clones (Supplementary Fig. 4). Notably, in AAG$^{-/-}$ELP1$^{-/-}$ cells, the co-regulated genes were significantly downregulated in comparison to WT levels, thus following the direction of ELP1$^{-/-}$ cells (Fig. 3g–i and Supplementary Fig. 3). This suggests that at the transcriptional level Elongator acts upstream of AAG. In summary, these results provide evidence of genes co-regulated

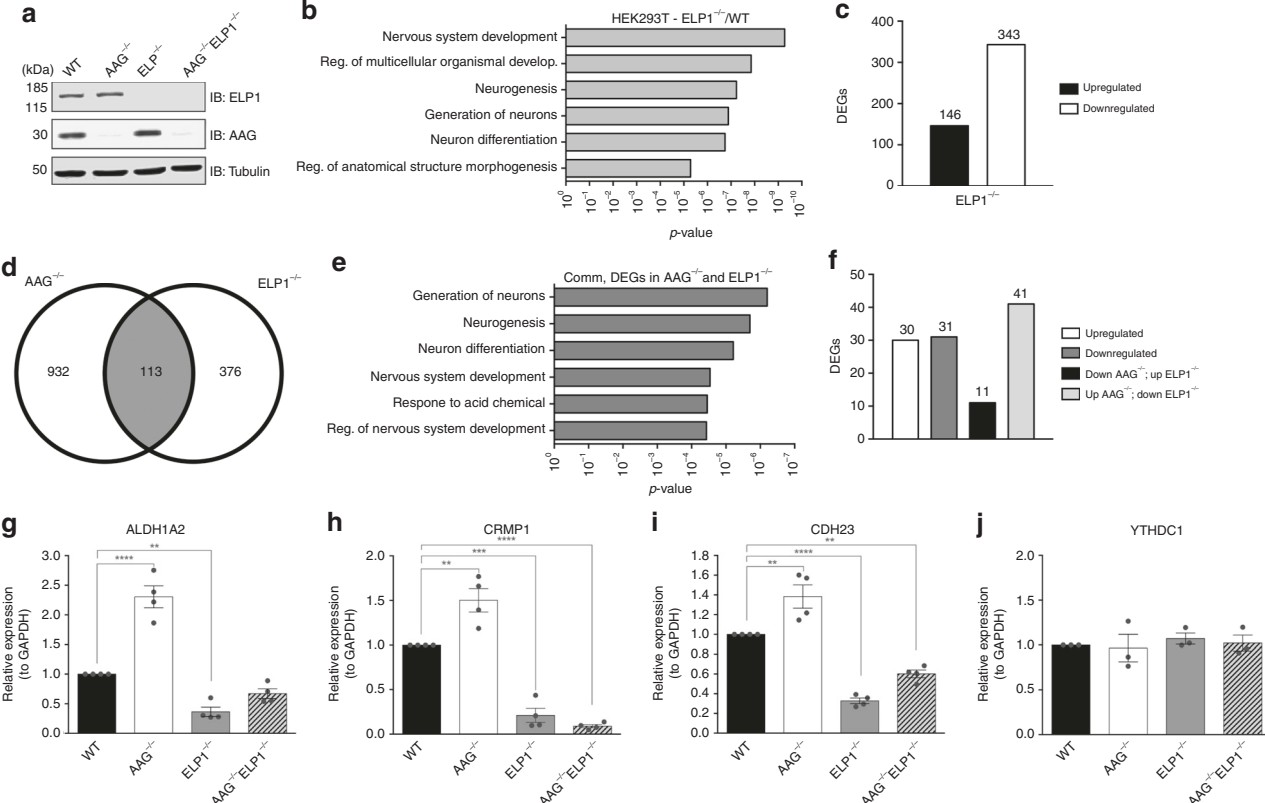

**Fig. 3** Elongator and AAG co-regulate gene expression. **a** Immunoblot of HEK293T whole-cell extracts (WCEs) from WT, AAG$^{-/-}$, ELP1$^{-/-}$, and AAG$^{-/-}$ELP1$^{-/-}$ cell lines generated via CRISPR-Cas9 technology. **b** Top six biological processes (BP) gene ontology (GO) terms as determined by DAVID for genes dysregulated in HEK293T ELP1$^{-/-}$ cells when compared to WT cells. **c** Downregulated and upregulated differentially expressed genes (DEGs) in HEK293T ELP1$^{-/-}$ cells. **d** Venn diagrams of AAG- and ELP1-regulated genes in HEK293T cells. **e** Top six BP GO terms as determined by DAVID of genes co-regulated by AAG and ELP1 in HEK293T cells. **f** Directionality of DEGs regulated by AAG and ELP1 in HEK293T cells. **g-j** Relative mRNA levels of *ALDH1A2* (**g**), *CRMP1* (**h**), *CDH23* (**i**) and *YTHDC1* (negative control) (**j**) genes in HEK293T WT, AAG$^{-/-}$, ELP1$^{-/-}$ and AAG$^{-/-}$ELP1$^{-/-}$ cells. Error bars represent mean ± SEM (*n* ≥ 3). **p ≤ 0.01; ***p ≤ 0.001; ****p ≤ 0.0001, one-way ANOVA. Source data are provided as Source Data file.

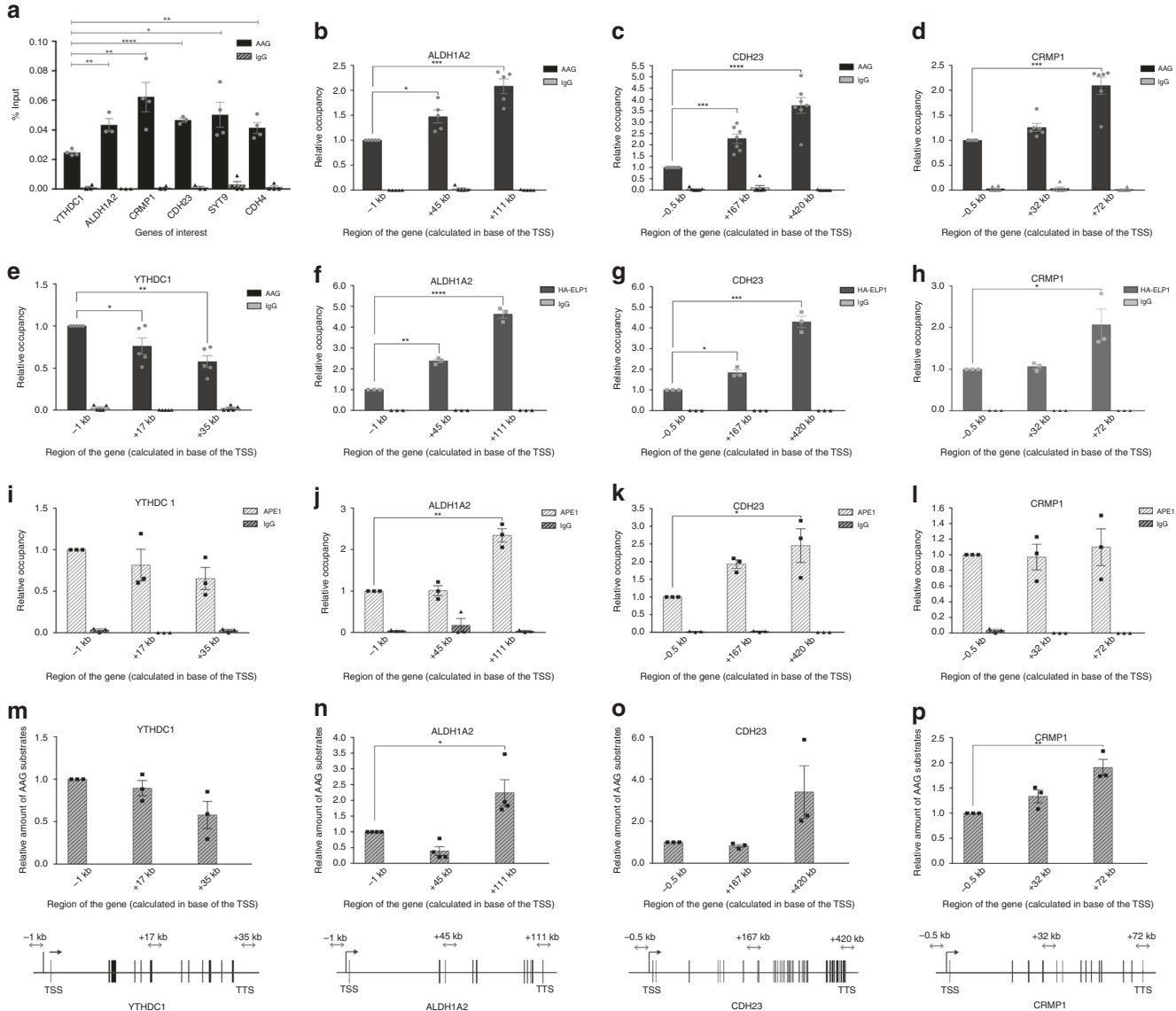

**Fig. 4** Elongator, components of AAG-initiated BER and AAG substrates accumulate towards the 3′end of co-regulated genes. **a** AAG ChIP-qPCR assays in HEK293T WT cells comparing percentage of input at gene bodies of unaffected gene (*YTHDC1*) and differentially expressed genes (*ALDH1A2, CRMP1, CDH23, SYT9,* and *CDH4*). **b–e** ChIP-qPCR assays showing relative AAG occupancy in AAG- and ELP1- dependent genes *ALDH1A2* (**b**), *CDH23* (**c**), *CRMP1* (**d**), and unaffected gene *YTHDC1* (**e**) in HEK293T WT cells. **f–h** ChIP-qPCR assays showing relative occupancy of HA-ELP1 in AAG- and ELP1- dependent *ALDH1A2* (**f**), *CDH23* (**g**), and *CRMP1* (**h**) genes in HEK293T HA-ELP1cells. **i–l** ChIP-qPCR assays showing relative APE1 occupancy in negative control *YTHDC1* gene (**i**), and AAG- and ELP1-dependent *ALDH1A2* (**j**), *CDH23* (**k**), *CRMP1* (**l**) genes in HEK293T WT cells. **m–p** qPCR DNA damage assay showing differences in distribution pattern of aberrant AAG substrates in unaffected gene Y*THDC1* (**m**) and genes regulated by AAG and ELP1: *ALDH1A2* (**n**), *CDH23* (**o**), *CRMP1* (**p**) in HEK293T WT cells. Values are shown as relative occupancy: % input of specific gene region, relative to percentage input of promoter region. Error bars indicate mean ± SEM (*n* ≥ 3). Two-tailed Student's *t*-test in **a**; one-way ANOVA in **b–p**; *$p \leq 0.05$, **$p \leq 0.01$, ***$p \leq 0.001$, ****$p \leq 0.0001$. Source data are provided as Source Data file.

by the AAG DNA repair glycosylase and its interaction partner the Elongator complex.

**AAG and Elongator accumulate in aberrant base-rich regions.** To determine whether AAG and ELP1 localize at the co-regulated genes identified in Fig. 2, chromatin immunoprecipitation (ChIP) experiments were performed. AAG ChIP showed significant enrichment at the co-regulated genes (*ALDH1A2*, *CRMP1*, *CDH23*, *SYT9*, and *CHD4*), when compared to *YTHDC1* unaffected control (Fig. 4a). Notably, AAG binding was most significantly increased towards the 3′end of the co-regulated genes (*ALDH1A2*, *CRMP1*, *CDH23*) (Fig. 4b–d). The defined 3′-end

enrichment was characteristic to co-regulated genes and was not present at *YTHDC1* (Fig. 4e). To determine the relation between ELP1 and AAG distribution, HEK293T cells with endogenously HA-tagged ELP1 were generated, using homologous recombination dependent CRISPR-Cas9 gene editing (Supplementary Fig. 5). Subsequent HA-ChIP experiments revealed that similar to the AAG distribution, and in line with its role in the transcription elongation, HA-ELP1 was significantly enriched towards the 3′ end of the co-regulated genes (Fig. 4f–h). Importantly, the same distribution pattern was observed for RNA pol II phosphorylated at Serine 2 of the C-terminal domain (CTD) (RNA pol II S2P), which is the predominant form during transcription elongation (Supplementary Fig. 6). Further, to test if other BER enzymes co-

occupy the same regions as AAG and ELP1, APE1 ChIP was performed. APE1 distribution closely resembled AAG and ELP1 localization at the co-regulated genes, with the highest levels detected towards the 3′end (Fig. 4i–l). Collectively, these results suggest that AAG-initiated BER associates with Elongator and transcription elongation, predominantly at the 3′end of the co-regulated genes. Since the main AAG function is to initiate BER, we next analyzed levels of aberrantly methylated AAG substrates along the co-regulated genes, using real-time qPCR based approach for quantification of aberrant DNA bases[35]. Interestingly, the distribution of endogenous aberrantly methylated AAG substrates closely followed AAG, APE1, ELP1 and RNA pol II pattern, with the levels of aberrant bases being highest towards the 3′end of the co-regulated genes (Fig. 4m–p). Taken together these findings suggest that regions of co-regulated genes that are co-occupied by AAG-initiated BER and Elongator have high levels of aberrant bases, thus indicating an interplay between the repair of DNA base lesions and transcription regulation.

**Loss of functional Elongator impairs AAG-initiated repair.** Since AAG interacts directly with ELP1 and the two proteins accumulate in the same gene regions, we next addressed the importance of ELP1 for the recruitment of AAG-initiated BER to

the chromatin. AAG ChIP in WT and ELP1$^{-/-}$ cells revealed that loss of ELP1 causes reduction in AAG binding to the chromatin at all genes tested (Fig. 5a–d), supporting that ELP1 is essential for AAG recruitment during transcription (Fig. 3g–j and Supplementary Fig. 3). Reduced AAG binding was not a consequence of perturbed chromatin organization in ELP1$^{-/-}$ cells, since histone occupancy was comparable in WT and ELP1$^{-/-}$ cells, as indicated by histone 3 ChIP (Supplementary Fig. 7a). Analysis of chromatin fractions isolated from WT and ELP1$^{-/-}$ cells demonstrated that ELP1 loss leads to global reduction of AAG on chromatin (Fig. 5e, f), while the total AAG levels were unchanged (Fig. 3a), further supporting the role of ELP1 in facilitating AAG recruitment. Similar to the impact on AAG localization, loss of ELP1 resulted in a notable reduction of APE1 recruitment to all tested genes (Fig. 5g). Since ELP1 promotes AAG chromatin recruitment, we next tested if ELP1 status could influence the complex formation between AAG and hyperphosphorylated RNA pol II. IP of AAG in WT and ELP1$^{-/-}$ HEK293T cells showed that loss of ELP1 diminishes the interaction between AAG and active RNA pol II S2P (Fig. 5h, compare lane 5 to 6). These findings suggest that ELP1 plays an important role in associating AAG with active transcription. To further evaluate the extent to which impaired AAG recruitment and reduced association with active transcription influence repair of aberrant bases in ELP1$^{-/-}$

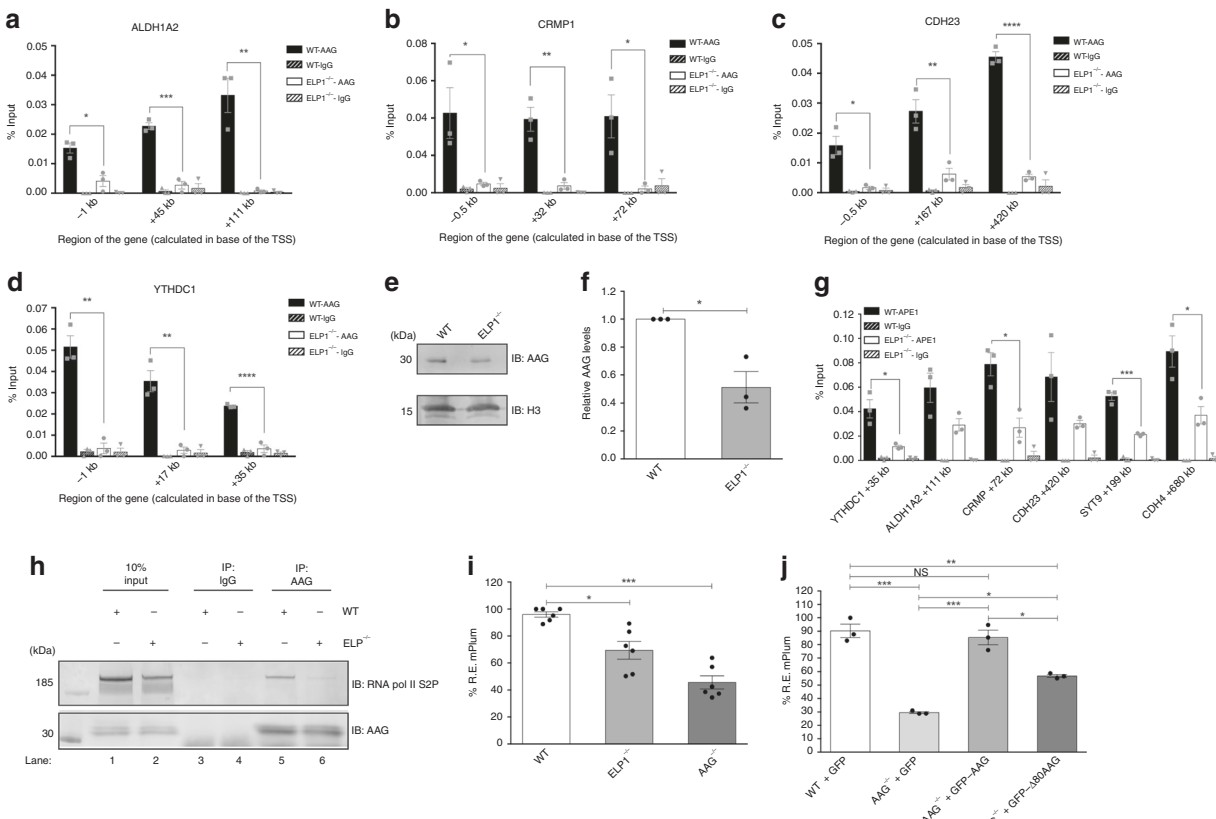

**Fig. 5** Lack of functional Elongator hampers AAG and APE1 chromatin recruitment, and AAG-initiated repair. **a–d** ChIP-qPCR experiments comparing AAG occupancy at promoters and gene bodies of *ALDH1A2* (**a**), *CRMP1* (**b**), *CDH23* (**c**), and *YTHDC1* (**d**) in HEK293T WT and ELP1$^{-/-}$ cells. **e** Immunoblot analysis of AAG levels in chromatin fraction of HEK293T WT and ELP1$^{-/-}$ cells. Histone H3 served as control. **f** Quantification of experiments as the one depicted in **e**. **g** ChIP-qPCR experiments comparing APE1 occupancy at gene bodies of *YTHDC1, ALDH1A2, CRMP1, CDH23, SYT9,* and *CDH4* in HEK293T WT and ELP1$^{-/-}$ cells. Error bars represent the SEM calculated from at least three independent experiments. **h** Immunoprecipitation of AAG from HEK293T WT and ELP1$^{-/-}$ whole cell extracts showing the interaction with RNA polymerase II phosphorylated at Serine 2 (S2P) of CTD repeat. **i** Measurement of AAG DNA glycosylase activity in HEK293T WT, AAG$^{-/-}$ and ELP1$^{-/-}$ on hypoxanthine (Hx)-containing plasmid by FM-HCR assays. **j** AAG DNA glycosylase activity determined by FM-HCR assays on hypoxanthine (Hx)-containing plasmid in HEK293T WT cells complemented with GFP and AAG$^{-/-}$ cells complemented with GFP, GFP-AAG or GFP-Δ80 AAG. Error bars represent mean ± SEM ($n \geq 3$). Two-tailed Student's t-test (**a–d, f, g**); one-way ANOVA (**i, j**), NS – non significant, *$p \leq 0.05$, **$p \leq 0.01$, ***$p \leq 0.001$, ****$p \leq 0.0001$. Source data are provided as Source Data file.

cells, AAG-initiated BER capacity was analyzed by flow cytometry host cell reactivation assay (FC-HCR)[36]. Importantly, cells lacking ELP1 showed significantly decreased capacity to repair AAG substrate hypoxanthine (Hx), when compared to WT cells (Fig. 5i). Hx was chosen over 3meA and 7meG, since these aberrantly methylated AAG substrates are unstable, and thus cannot be readily incorporated at the specific site in the DNA repair construct. While our findings suggest that global loss of ELP1 impairs repair (Fig. 5i), the necessity of ELP1 binding for efficient AAG-initiated BER remains unclear. To address this we compared BER capacity by FC-HCR analysis, in AAG$^{-/-}$ cells complemented with full-length or Δ80 AAG, lacking the unstructured region important for ELP1 binding (Fig. 2). Importantly, previous work showed that the catalytic activities of full-length AAG and Δ80 AAG are very similar[37,38]. Both full-length and Δ80 AAG were GFP-tagged, allowing to directly relate BER capacity to the amount of protein present in the cells, and were expressed at level comparable to endogenous AAG in WT cells (Supplementary Fig. 7b). Interestingly, while complementation of AAG$^{-/-}$ cells with GFP-AAG successfully rescued repair capacity, the AAG-initiated BER remained significantly reduced in AAG$^{-/-}$ cells expressing GFP-Δ80, when compared to WT cells (Fig. 5j). Taken together, our findings suggest that ELP1 binding as well as functional Elongator complex, are prerequisite for effective AAG chromatin recruitment, association with actively transcribing RNA pol II and AAG-initiated BER.

**Active transcription promotes repair of aberrant bases.** Our findings suggest that AAG-initiated BER is associated with transcription through direct interaction with ELP1, and that Elongator promotes AAG-initiated BER in actively transcribing genes (Figs. 2 and 5). It remains however unclear to which extent active transcription directly influences AAG occupancy, and what is its importance in the repair of aberrant bases. To determine if transcription inhibition affects AAG distribution, ChIP experiments were performed in untreated cells and cells exposed to 5,6-dichloro-1-beta-D-ribofuranosylbenzimidazole (DRB). Treatment with transcription inhibitor DRB results in repression of transcription elongation, as consequence of impaired RNA pol II CTD phosphorylation at Serine 2 (refs[39,40]), thus causing reduced

RNA pol II occupancy (Supplementary Fig. 8). Similar to the ELP1 loss (Fig. 5), transcription inhibition resulted in globally reduced AAG binding at all analyzed chromatin regions in DRB treated cells (Fig. 6a–d). This further suggests that transcription is an important modulator of AAG chromatin occupancy. To next determine if active transcription, besides promoting AAG occupancy, also has a role in AAG-initiated repair a single-cell AAG-Comet FLARE analysis was performed. Interestingly, inhibition of transcription elongation caused significant accumulation of AAG-specific aberrant bases, while it did not affect global DNA damage levels (Fig. 6e). In summary, these results strongly suggest that active transcription is required to promote AAG recruitment to the chromatin and to facilitate AAG-initiated BER of aberrant bases.

## Discussion

Removal and repair of aberrant bases are dramatically impaired in the context of chromatinized DNA[9–12,41]. It has been suggested that for efficient repair within chromatin to take place, BER needs to be associated with essential nuclear processes, such as transcription[6]. Besides the potential importance of transcription in promoting efficient BER, several studies indicated that BER enzymes, in particular DNA glycosylases, could influence transcription and play an important role in modulation of gene expression[19–21,42]. However, direct evidence for the existence of transcription associated BER has not been provided so far. In this work we show that AAG-initiated BER associates with the transcription machinery, primarily by binding to the Elongator complex (Fig. 2). Set of IP experiments indicates that AAG forms a complex with active RNA pol II (Fig. 1c), predominantly through direct association of its unstructured N-terminal region with the ELP1 subunit of the Elongator (Figs. 2 and 5h). Identification of direct interaction between AAG and the ELP1 confirms recent predictions arising from a large proteomics screen[22]. Results of PLA experiments further indicate that this interaction primarily takes place in the nucleus (Fig. 2g). The relevance of AAG interaction with Elongator is demonstrated by the RNA sequencing, presented in Fig. 3d, which revealed that AAG and ELP1 co-regulate a specific set of genes. On the chromatin level, the occupancy of the BER components AAG and APE1 accompanies the progressive enrichment of ELP1 and elongating RNA pol II at

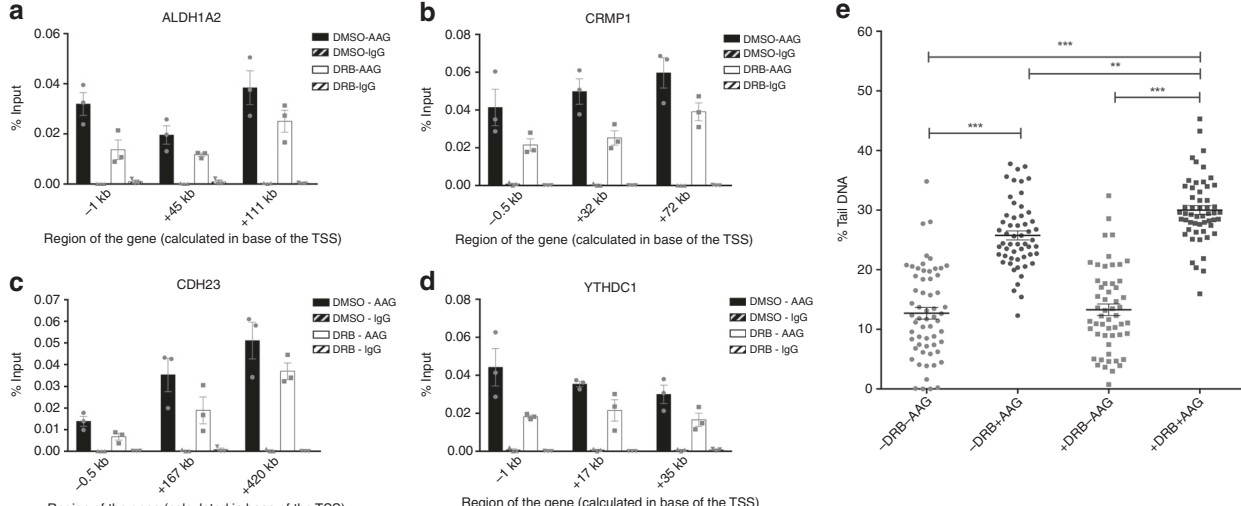

**Fig. 6** Transcription inhibition reduces AAG occupancy and impairs removal of AAG substrates. **a–d** ChIP-qPCR experiments comparing AAG occupancy in DMSO and DRB treated HEK293T WT cells at *ALDH1A2* (**a**), *CRMP1* (**b**), *CDH23* (**c**), and *YTHDC1* (**d**) genes. Error bars, SEM from three independent experiments. **e** AAG Comet-FLARE assay in DMSO or DRB treated HEK293T cells. Error bars indicate mean ± SEM ($n \geq 3$). **$p \leq 0.01$; ***$p \leq 0.001$, one-way ANOVA. Source data are provided as Source Data file.

the 3′end of the co-regulated genes (Fig. 4). The ELP1 and RNA pol II distribution patterns are directly in line with the previous observations[30]. Importantly, regions of the co-regulated genes, enriched for AAG, APE1, and ELP1 present with the highest level of endogenous aberrantly methylated AAG substrates (Fig. 4m–p). Thus suggesting potential role of aberrantly methylated bases in regulation of gene expression. This is particularly interesting in light of recent work, which indicated coevolution of epigenetic DNA methylation with repair of aberrantly methylated DNA bases, including AAG-initiated BER[43]. Further, accumulation of aberrantly methylated AAG substrates coincides with accumulation of elongating RNA pol II S2P (Supplementary Fig. 6), which was shown to be indicative of a reduced elongation rate due to increased nucleosome density[44]. Indeed, the findings provided by genome-wide mapping of aberrantly methylated bases in yeast, suggested that repair of AAG substrates is slower in the regions with strongly positioned nucleosomes, in comparison to the nucleosome depleted regions[45].

As in the case of RNA pol II complex formation, the recruitment of AAG-initiated BER to the chromatin is strongly dependent on functional Elongator, and results in dramatic decrease of AAG and APE1 occupancy, both in specific gene regions, as well as in chromatin bound fraction of ELP1$^{-/-}$ cells (Fig. 5). Besides impact on the chromatin recruitment, analysis of AAG-repair capacity revealed that ELP1 binding is necessary for efficient AAG-initiated repair (Fig. 5i, j). Accordingly, both loss of functional Elongator in ELP1$^{-/-}$ cells, as well as lack of the ELP1 binding region within AAG (Δ80 AAG), resulted in significantly impaired AAG-initiated repair. In line with Elongator importance for efficient repair, AAG-initiated BER is further directly dependent on active transcription as indicated by experiments involving transcription elongation inhibitor DRB. Notably, DRB treatment resulted in reduced AAG occupancy and impaired AAG-initiated repair (Fig. 6), thus suggesting that active transcription has an important role in the maintenance of genome stability by facilitating BER. Very recently Menoni et al. similarly indicated that the inhibition of transcription elongation by DRB results in reduced recruitment of the BER scaffold protein XRCC1 to the sites of initiated repair[46]. Interestingly, early work in mouse cells did not detect impact of transcription coupled repair in clearance of methylation damage[47]. However this study largely focused on the removal of aberrantly methylated bases from a single specific gene. The fact that inhibition of transcription elongation impairs AAG-initiated BER and results in global accumulation of aberrantly methylated AAG substrates, strongly supports the idea that one key functional consequence of Elongator-mediated AAG association with the active transcription is ensuring efficient repair. Accumulation of AAG substrates genome wide could directly have harmful effects on transcription, since 3meA was shown to affect the fidelity of incorporation by RNA pol II[48]. While the 3meA analog 3-deaza-3meA can successfully be bypassed by RNA pol II, its presence causes 10-fold greater misincorporation of CMP, thus resulting in synthesis of mRNA's bearing mutations. In contrast to 3meA, AP site intermediates, generated by APE1 during BER, impair RNA pol II progression[49]. Finding that AAG, similarly to other DNA glycosylases[42], primarily inhibits gene expression (Fig. 1g) suggests that by coordinating BER initiation with transcription, AAG could potentially inhibit RNA pol II progression and therefore repress generation of faulty transcripts. Imbalanced AAG-initiated BER and consequent accumulation of aberrantly methylated bases can thus, besides causing genome instability and increased mutation rate, also impact gene expression.

Loss of AAG results in increased expression of genes that primarily segregate in neurodevelopmental processes (Fig. 1 and Supplementary Fig. 1). It has been demonstrated recently that the DNA glycosylases Ogg1 and Mutyh, involved in repair of oxidized bases, play an important role in neurodevelopment through impact on transcription. Lack of Ogg1 and Mutyh was reported to upregulate ERα target-genes, which in turn modulated cognition and anxiety-like behavior in mice[21]. It will be thus interesting to determine the importance of transcription associated AAG-initiated BER in regulation of neurodevelopment, and test its role in brain functioning.

Taken together, based on our findings we propose a model of AAG-initiated DNA repair coordinated with gene expression (Fig. 7). During transcription Elongator associates with RNA pol II and promotes transcription elongation, accumulating towards the 3′end of regulated genes. Through its unstructured N-terminal region AAG associates with the ELP1 subunit of Elongator and forms a complex with the active transcription machinery. As a consequence of active transcription chromatin is suggested to be locally decondensed, which allows AAG to efficiently initiate BER by recognizing and removing aberrant bases. BER initiation likely temporarily inhibits RNA pol II progression, resulting in reduced expression of co-regulated genes. In the absence of Elongator, transcription of target genes is impaired due to reduced elongation, while AAG chromatin recruitment and BER initiation are hampered. In summary our results suggest that the transcriptional status, in addition to the type of aberrant bases and DNA glycosylases involved, is an essential determinant of BER efficiency and its impact on genome and chromatin organization. Association of AAG with transcription elongation could thus, in addition to enabling efficient removal of aberrantly methylated bases, serve as an important layer of gene expression control.

## Methods

**Cells and culturing**. Human embryonic kidney cells expressing a mutant version of the SV40 large T antigen (HEK293T, American Tissue Culture Collection, CRL-3216™) were maintained under 5% CO$_2$ and 37 °C in DMEM high glucose supplemented with 10% fetal bovine serum, and 1% penicillin/streptomycin. HAP1 cells knockout and parental WT (Horizon Genomics, HZGHC001537c002 and HZGHC001537c003) were maintained under 5% CO$_2$ and 37 °C in IMDM supplemented with 10% fetal bovine serum, 1% penicillin/streptomycin, and 1% L-Glutamine (L-Glut).

**Fractionation**. To obtain total fraction (TF) $2.5 \times 10^6$ HEK293T cells were resuspended in 200 μL of MNase buffer (50 mM Tris-HCl pH 7.5, 30 mM KCl, 7.5 mM NaCl, 4 mM MgCl$_2$, 1 mM CaCl$_2$, 0.125% (v/v) NP-40, 0.25% Na-deoxycholate, 0.3 M sucrose, 1x Halt™ Protease Inhibitor Cocktail) with 1U MNase, and the samples incubated at 37 °C for 30 min. For the preparation of the soluble fraction (SF) and chromatin fraction (CF) $2.5 \times 10^6$ cells were resuspended in 200 μL of chromatin extraction buffer (10 mM HEPES pH 7.6, 3 mM MgCl$_2$, 1 mM DTT, 1x Halt™ Protease Inhibitor Cocktail) and samples rotated 30 min at RT. SF was separated from nuclei by low-speed centrifugation (1300 × g for 10 min) and collected for further analysis. Pelleted nuclei were lysed in 200 μL of MNase buffer with 1U MNase, for 30 min at 37 °C. TF, SF, and CF were boiled in Laemmli buffer, centrifuged at 16,000 × g for 5 min at RT, and equal volumes of each fraction subjected to immmunoblott analysis.

**Generation of HEK293T knockout and HA-ELP1 cell lines**. CRISPR short guide RNAs (sgRNAs) were designed using the Optimized CRISPR Design tool (http://tools.genome-engineering.org). The oligo pairs encoding the sgRNAs (Supplementary Table 1) were annealed, ligated into pSpCas9(BB)-2A-GFP (PX458) (Addgene plasmid # 48138; a gift from Feng Zhang) at BbsI site. Next, HEK293T WT cells were transiently transfected using calcium phosphate and seeded as single clones into a 96-well plate via several dilution. Mutations and loss of target protein expression was confirmed in the clonal KO cell lines by sequencing and immunoblot analysis, respectively. The presence of off-target mutations in KO cells was excluded by sequencing of top-candidate off-target sites predicted by Optimized CRISPR Design tool. For generation of HEK293T HA-ELP1 cells plasmid containing the sgRNA for ELP1 C-terminus was transfected together with the HA-ELP1 repair oligo containing BspEI specific cut-site (Supplementary Table 1) into HEK293T cells using nucleofector (Lonza), according to the manufacturer's protocol. The cells were tested for HA insertion by: PCR followed by specific BspEI digestion, sequencing, and immunoblot analysis.

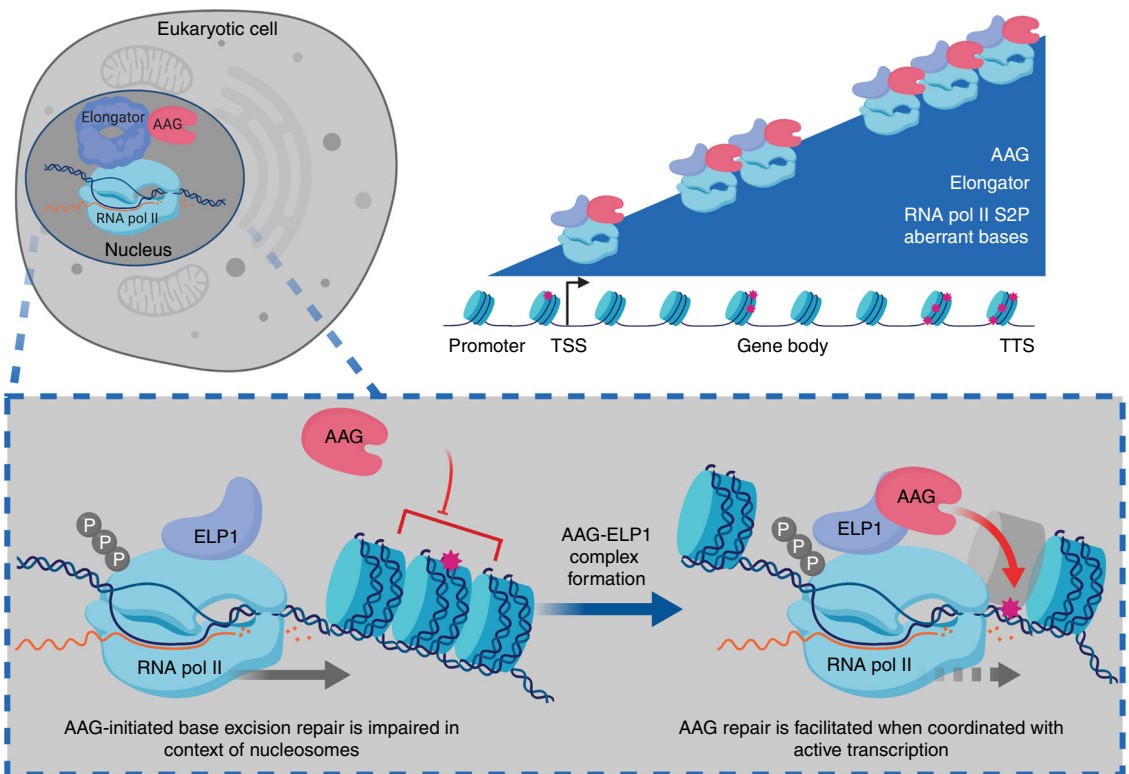

**Fig. 7** Model of AAG-initiated DNA repair coordinated with gene expression. Elongator complex associates with RNA pol II to promote transcription elongation, accumulating towards 3′end of regulated genes. AAG through its unstructured N-terminal region associates with ELP1 subunit of Elongator, thus forming complex with active transcription machinery. During active transcription chromatin is locally decondensed, which allows AAG to efficiently initiate BER by recognizing and removing aberrant bases. AAG-initiated BER likely temporarily inhibits RNA pol II progression, thus resulting in reduced expression of co-regulated genes. In the absence of Elongator, transcription of target genes is repressed, while AAG recruitment to chromatin and initiation of BER is impaired. For more details see text. Schematic representation was created with Biorender.com.

**RNA isolation, library preparation, and RNA sequencing**. Total RNA was purified with RNeasy kit (QIAGEN), and DNaseI digested (QIAGEN) according to the manufacturer's protocol. RNA library preparation, and sequencing of HEK293T samples, were carried out by the Functional Genomic Centre, Zurich (FGCZ). The libraries were screened on the Bioanalyzer (Agilent), pooled in equimolar concentrations, and sequenced on one lane of a flow cell on an Illumina HiSeq2000 using single-read TruSeq v3 chemistry with 125-cycles. PolyA RNA sequencing of HAP1 was performed by the Beijing Genomics Institute (BGI). Briefly, polyA RNA was isolated and generated into single read, 50 bp libraries following standard Complete Genomics protocols.

**Bioinformatic processing**. FASTQ files of HEK293T samples were processed to remove adapter sequences and low quality reads prior to alignment to the GRCh37 Ensembl release 80 of the human genome using STAR v2.4.2a aligner[50]. Reads within exon features were counted using featureCounts in Bioconductor (3.2) package Rsubread (1.20.6). Differential expression analysis was performed in R using the Bioconductor package DESeq2 (ref. [51]). Pairwise comparisons were made between the HEK293T WT and the knockout cell-lines, AAG$^{-/-}$ or ELP1$^{-/-}$ samples. For the WT vs AAG$^{-/-}$ comparisons, statistically significant differentially expressed genes were defined as having a ≥1.5-fold change and FDR ≤ 0.1. For the WT vs ELP1$^{-/-}$ comparisons, statistically significant differentially expressed genes were defined as having a ≥2-fold change and FDR ≤ 0.1. DAVID v6.8[32] gene-ontology (GO) enrichment analysis was performed to identify enriched Biological Processes (BP).

FASTQ files from HAP1 samples were processed to remove adapter sequences and low quality reads prior to alignment to the hg18 genome. Specifically, reads in which mismatched bases are more than 10% and low quality reads (the percentage of low quality bases is over 50% in a read) are removed using a custom BGI pipeline. Reads passing filtering were mapped to the hg18 reference transcriptome using Bowtie2 (ref. [52]) and BWA[53] was used for mapping to the reference genome. Gene Quantification done using RSEM[54], which calculates FPKM. DEGs screened using Poisson Distribution Method, performed using custom BGI pipelines.

**Whole cell extracts**. HEK293T cells were trypsinized and washed twice with ice-cold PBS and the pellet flash frozen in liquid $N_2$. The cell pellets were next resuspended in 2 PCV of hypotonic lysis buffer (20 mM Hepes pH 7.9, 2 mM MgCl, 0.2 mM EGTA, 10% glycerol, 0.1 mM PMSF, 2 mM DTT, 1x Halt™ Protease

Inhibitor Cocktail), left on ice for 5 min and subjected to three freeze and thaw cycles. Subsequently, NaCl and NP-40 were added to final concentration 0.5 M and 0.5% (v/v) respectively, and samples incubated 20 min on ice. The samples were diluted by addition of 8 PCV hypotonic lysis buffer containing 50 mM NaCl and subjected to sonication. In case of nuclease treatment either 1.2 μl CaCl$_2$ (2.5 M) with 25U of DNase, or 2.1 μg RNaseI, and 50U MNase and samples incubated for 1 h, at 4 °C. Samples were centrifuged and the WCE supernatant collected.

**Protein mass spectrometry**. Using calcium phosphate precipitation empty pcDNA3.1-FLAG or pcDNA3.1-FLAG-AAG expression constructs (0.1 μg ml$^{-1}$) were transiently transfected into HEK293T cells. Before harvesting, cells were mock or MMS (2 mM) treated for 1 h and WCEs prepared. Next, 1 mg WCE proteins was incubated with 20 μl of FLAG M2 resin (Sigma) in buffer A1 (20 mM HEPES at pH 7.9, 2 mM MgCl$_2$, 0.2 mM EGTA, 10% (v/v) glycerol, 0.1 mM PMSF, 2 mM DTT, 140 mM NaCl, 0.01% (v/v) Nonidet-P40) for 2 h at 4 °C. The resin was washed four times using the same buffer, bound samples eluted in two steps with buffer A1 containing 100 μg ml$^{-1}$ 3 × FLAG peptide (Sigma), and subjected to trichloroacetic acid precipitation. Identification of proteins in FLAG-AAG mock or MMS treated, and control FLAG samples was performed by the Biopolymers and Proteomics Core Facility of the David H. Koch Institute for Integrative Cancer Research. Briefly, protein samples were reduced, alkylated, and digested with trypsin, followed by purification and desalting on an analytical C18 column tip. The processed peptides were then analyzed on an Agilent Model 1100 Nanoflow high-pressure liquid chromatography (HPLC) system coupled with electrospray ionization on a Thermo Electron Model LTQ Ion Trap mass spectrometer (MS). All MS/MS samples were analyzed using Sequest (Thermo Fisher Scientific, San Jose, CA, USA; version SRF v. 3). Sequest was set up to search the merged_human_90T.fasta.hdr database assuming the digestion enzyme trypsin. Sequest was searched with a fragment ion mass tolerance of 0.50 Da and a parent ion tolerance of 2.0 Da. Dehydro of serine and threonine, oxidation of methionine, iodoacetamide derivative of cysteine and phosphorylation of serine, threonine and tyrosine were specified in Sequest as variable modifications. Scaffold (version Scaffold_4.3.0, Proteome Software Inc., Portland, OR) was used to validate MS/MS based peptide and protein identifications. Peptide identifications were accepted, if they could be established at >95.0% probability (as specified by the Peptide Prophet algorithm[55]). Peptide identifications were also required to exceed specific database search engine thresholds. Sequest identifications required at least deltaCn scores of greater than

0.10 and XCorr scores of greater than 2.0, 2.5, 3.5, and 3.5 for singly, doubly, triply and quadruply charged peptides. Protein identifications were accepted if they could be established at greater than 99.0% probability and contained at least 2 identified peptides. Protein probabilities were assigned by the Protein Prophet algorithm[56]. Proteins that contained similar peptides and could not be differentiated based on MS/MS analysis alone were grouped to satisfy the principles of parsimony. All proteins identified in FLAG-AAG and control Flag samples, of cells untreated or treated with MMS are indicated in Supplementary Data 2 and available in PRIDE under accession PXD013508.

**Co-immunoprecipitation (CoIP) assays.** WCEs (0.5 mg) were incubated with 2 μg of antibody (anti-AAG antibody;[8] or anti-ELP1, Abcam ab56362) or IgG (rabbit IgG, Diagenode, C15410206; or mouse IgG, Diagenode, C15400001) at 4 °C overnight. In case of purified proteins (FLAG-ELP1, AAG or Δ80 AAG) 1 μg of protein was mixed with 1 μg of antibody and incubated overnight at 4 °C in 200 μl final volume of IP buffer (20 mM HEPES pH 7.9, 2 mM MgCl$_2$, 0.2 mM EGTA, 10% (v/v) glycerol, 0.1 mM PMSF, 2 mM DTT, 140 mM NaCl, 0.01% (v/v) Nonidet-P40). Protein-A Dynabeads, or Protein-A sepharose beads (in case of purified proteins), were equilibrated in the IP buffer at 4 °C overnight. After three consecutive washes the beads were added to samples and incubated for 4 h at 4 °C. The supernatant was removed and beads washed three times with cold wash buffer. Beads were boiled in Laemmli buffer and subjected to immunoblot analysis.

**Immunoblot analysis.** Samples were separated on 4–12% Bis–Tris polyacrylamide gel (Invitrogen) followed by transfer to Amersham™ Hybond® P 0.2 PVDF (GE Healthcare) for immunoblotting. Primary antibodies: anti- anti-ELP1 (1:250, Abcam, ab56362), anti-ELP3 (1:1000, Abcam, ab96781), anti-AAG (1:1000, custom rabbit polyclonal antibody, Covance, raised against Δ80AAG) or anti-AAG (1:1000, LSBio LS-C133325), anti-RNA pol II S2P (1:1000, Abcam, ab5095), anti-HA (1:1000, Abcam, ab9110), anti-α-Tubulin (1:10000, Cell Signaling, 2144), anti-H3 (1:1000, Abcam, ab1791), anti-GFP (1:1000, Abcam, ab290); were detected using infrared (IR) Dye-conjugated secondary antibodies (1:15000, Li-COR Biosciencecs, 827-11081 and 925-32210). The signal was visualized by using direct IR fluorescence via the Odyssey Scanner, LI-COR Biosciences.

**Column purification of cellular complexes.** To isolate cellular complexes 1 ml, HeLa cell pellet was washed twice with 1 packed cell volume (PCV) of ice-cold PBS, and resuspended in 4PCV of hypotonic buffer (10 mM Tris-HCl pH 8.0, 1 mM EDTA, 5 mM DTT) and incubated at 0 °C for 20 min. After addition of 0.5 mM PMSF and 0.5 μg ml⁻¹ Leupeptin, Pepstatin, Chymostatin, cells were homogenized using a Douncer and 4PCV of ice-cold buffer 2 (50 mM Tris-HCl pH 8.0, 10 mM MgCl$_2$, 2 mM DTT, 25% sucrose, 50% (v/v) glycerol) were slowly added while stirring. Next, 1 PCV of neutralized saturated ammonium sulfate solution was added during stirring for 30 min at 0 °C, followed by ultracentrifugation 3 h at 2 °C, 45'000 rpm. The supernatant was collected and dialyzed into buffer B (20 mM Tris Acetate pH 7.8, 20% (v/v) glycerol, 1 mM EDTA, 1 mM DTT, 0.01% (v/v) NP-0.4, 1 mM PMSF, 2 μM Pepstatin A, 2 μg ml⁻¹ Chymostatin, 0.6 μM Leupeptin, 2 mM Benzamidine) with 50 mM KCl; 3 times for 1 h at 4 °C. Dialyzed samples were load onto heparin-sepharose column overnight. After washing with 10 column volumes of buffer B supplemented with 50 mM KCl, bound proteins were eluted as 1 ml fractions using buffer B with 150 mM, 300 mM, 450 mM, 700 mM and 1000 mM KCl, sequentially. The different elutions were subjected to immunoblot analysis.

**Protein purification.** Human full length AAG and Δ80AAG lacking first 80 N-terminal amino acids were expressed and purified by using the established expression system[57]. ELP1 cDNA was excised from pCMV6-XL5-ELP1 (OriGene) by NotI (NEB) and inserted into pcDNA3.1(+)-3xFLAG vector generated as described in[8]. HEK293T cells were transfected by calcium phosphate, harvested after 48 h and WCE prepared as described above. 0.5 mg WCE were incubated with 75 μl anti-FLAG M2 affinity beads (Sigma Aldrich) for 2 h at 4 °C, the beads were washed three times with washing buffer (20 mM Hepes pH 7.9, 2 mM MgCl$_2$, 0.2 mM EGTA, 10% (v/v) glycerol, 0.1 mM PMSF, 2 mM DTT, 140 mM NaCl, 1x Halt™ Protease Inhibitor Cocktail). FLAG-ELP1 was eluted twice by addition of 80 μl Elution buffer (washing buffer with 0.5% (v/v) NP-40 and 0.15 μg μl⁻¹ 3x FLAG peptide (Sigma Aldrich)) for 30 min. The supernatant was applied on an Amicon Ultra-0.5 Centrifugal Filter Unit (Merck) and buffer exchanged to storage buffer (washing buffer with 0.1% (v/v) NP-40).

**Gel shift experiments combined with chemical crosslinking.** To evaluate affinity between AAG and ELP1 10 μM of purified proteins were mixed in a final volume of 50 μl reaction buffer (0.1 M sodium phosphate, 0.15 M NaCl, pH 7.4) and equilibrated 30 min at RT. Crosslinking agent BS³ (Thermo Fisher Scientific) was next added in different amounts: 0 (control), 0.1 mM and 0.5 mM and samples incubated 1 h at 4 °C, shaking. The reactions were quenched with 2.5 μl of 1 M Tris (pH 7.5) during 5 min at RT. As an additional control, samples containing single proteins alone were also evaluated in the presence of BS³. Samples were analyzed via immunoblot analysis with anti-AAG (1:1000, LSbio, LS-C133325), anti-ELP1 (1:1000, Genosphere custom antibody, raised against 1199-1218aa of ELP1 NP_0013052891) and appropriate secondary antibodies.

**GFP-AAG and AAG-GFP constructs.** The cDNA of AAG transcript variant 1 (NM_002434.4) was inserted into the pEGFP-C1 or pEGFP-N1 (Clontech) using BamHI, XbaI and NotI, XbaI respectively, to create pEGFP-C1-AAG and pEGFP-N1-AAG, which were validated by restriction enzyme digestion and sequencing. To generate AAG truncated forms: containing first 80 N-terminal AAG amino acids (1-80aa AAG) or AAG lacking 80 N-terminal amino acids (Δ80 AAG), pEGFP-C1-AAG plasmid was subjected to site directed mutagenesis using specific primer sets (Supplementary Table 2). Generated pEGFP-C1-(1-80aa AAG), pEGFP-N1-(1-80aa) and pEGFP-C1-(Δ80 AAG) were confirmed by sequencing.

**GFP-Trap co-immunoprecipitation.** In all, $0.3 \times 10^7$ HEK293T cells were transfected by calcium phosphate, harvested after 48 h, resuspended in 200 μl ice-cold lysis buffer (10 mM Tris HCl pH 7.5, 150 mM NaCl, 0.5 mM EDTA 0.5% NP-40, 1 mM PMSF, 1X protease inhibitors) and incubated at 4 °C for 30 min. WCE was isolated as supernatant after centrifugation $20.000 \times g$ for 10 min at 4 °C. 25 μL GFP-Trap®_MA (Chromotek) beads prewashed in wash buffer (10 mM Tris HCl pH 7.5, 150 mM NaCl, 0.5 mM EDTA) was added to 0.5 mg of WCE and the mixture was incubated for 2 h at 4 °C with constant rotation. The beads were washed three times with wash buffer. Proteins were eluted off the beads by boiling the beads in Laemmli sample buffer for 10 min and subjected to SDS-PAGE electrophoresis and immunoblot analysis.

**Duolinkproximity ligation assay.** To visualize close proximity between AAG and the ELP1, HEK293T cells were fixed in 4% paraformaldehyde for 15 min at RT, washed three times with PBS and permeabilized with PBS containing 0.1% Triton X-100 for 40 min at RT in Poly-D-Lysine and laminin double-coated plates. PLA was performed using Duolink® detection kit according to manufacturer's instructions (Sigma-Aldrich) for analysis of dual PLA receptor recognition (AAG-ELP1 proximity) using AAG (1:80, LSBio LS-C133325) and ELP1 (1:50, Genosphere custom rabbit polyclonal antibody, raised against 1199-1218aa of ELP1 NP_0013052891) primary antibodies. Cells were washed, and the cell nuclei were counterstained using DAPI. Z-stacks were collected by Zeiss LSM 880 confocal microscope, with an Plan-Apochromat 63 × /1.4 oil DICM27 objective and ZEN v2.5 software. The images were obtained as a compilation of confocal Z-stacks comprising 15 optical spices (0.5 μm intervals) into 2D using maximum intensity projection using ImageJ.

**Gene expression analysis.** RNA was purified with RNeasy kit (QIAGEN) according to the manufacturer's protocol and reverse transcribed using MultiScribe Reverse Transcriptase (QIAGEN) according to the manufacturers protocol. qPCR was performed with Power SYBR Green PCR Master Mix (Applied Biosystems) on a StepOnePlus v2.3 Real-Time PCR System. Relative transcription levels were determined by normalizing to GAPDH mRNA levels. All primer sequences are listed in Supplementary Table 3.

**Chromatin immunoprecipitation.** HEK293T cells were crosslinked with 1% formaldehyde and quenched with 0.110 mM glycine. Cells were washed with ice-cold PBS, and harvested. Cell pellets were resuspended in cell lysis buffer (100 mM Tris-HCl pH 8, 10 mM DTT) and incubated 15 min on ice, and 15 min at 30 °C shacking. Next, nuclei were pelleted and washed with buffer A (10 mM EDTA pH 8, 10 mM EGTA, 10 mM HEPES pH 8, 0.25% Triton X-100), followed by buffer B (10 mM EDTA pH 8, 0.5 mM EGTA, 10 mM HEPES pH 8, 200 mM NaCl). Nuclei were lysed in lysis buffer (50 mM Tris-HCl pH 8, 10 mM EDTA and 1% SDS) and chromatin sheered to 200-250 bp DNA fragments by sonication with Bioruptor (Diagenode) for 30 cycles of 30 s. In all, 40 μg of chromatin was next precleared 2 h at 4 °C, and then incubated with 2 μg antibody in ChIP buffer (16.7 mM Tris-HCl pH 8, 167 mM NaCl, 1.2 mM EDTA, 0.01% SDS, and 1.1% Triton X-100) overnight at 4 °C. ChIP antibodies included: anti-AAG (LSBio, LS-C133325-100); ant-HA (Abcam, ab9110); anti-APE1 (Abcam, ab194); anti-RNA pol II S2P (Abcam, ab5095); anti-H3 (Abcam, ab1791); anti-RNA pol II (MBL, MABI0601). The DNA-protein-antibody complexes were isolated using A/G dynabeads (Thermofischer) and washed using sequentially: low salt wash buffer (16.7 mM Tris-HCl pH 8, 167 mM NaCl, 0.1% SDS, 1% Triton X), high salt wash buffer (16.7 mM Tris-HCl pH 8, 500 mM NaCl 0.1% SDS, 1% Triton X) and LiCl wash buffer (250 mM LiCl, 0.5% NP40, 0.5% Na-deoxycholate, 1 mM EDTA, 10 mM Tris-HCl pH 8). Proteinase K treatment was performed for 1 h at 50 °C with 10 mM EDTA, 40 mM Tris-HCl pH 6.5 and 20 μg proteinase K. The DNA was purified with phenol-chloroform, ethanol precipitated and analyzed by qPCR. Levels of ChIPed DNA is expressed as percentage of input, or relative abundance = (% input of specific gene region)/(% input of promoter region). Primer sequences are listed in the Supplementary Table 4.

**Region-specific quantification of aberrant AAG substrates.** The levels of aberrant bases substrates of AAG were determined in sheared HEK293T WT genomic DNA with an average fragment size of 200 bp. DNA was incubated with either a combination of 10U of AAG (NEB, M0313) and 10U of APE1 (NEB, M0282), or only with 10U of APE1 for 1 h at 37 °C in 1X ThermoPol Reaction Buffer (NEB, B9004). Next, level of DNA damage was assessed by qPCR (StepOne Software version v2.3) using primers targeting regions corresponding to promoter, middle and end of the gene of interest (Supplementary Table 4). Level of AAG

substrates in the targeted regions was inferred by calculating: $\Delta Ct = Ct( + AAG + APE1) - Ct(-AAG + APE1)$. The relative amount of AAG substrates for each gene region was calculated with respect to the promoter.

**Flow cytometry host cell reactivation assay.** Flow cytometry host cell reactivation assay (FM-HCR) assay was performed using plasmids for expression of the fluorescent proteins EGFP and mPlum subcloned into the pmaxCloning. EGFP plasmid served as control. The repair mPlum plasmid (mPlum.Hx) was engineered by placing the AAG-specific base lesion hypoxanthine (Hx) lesion into the open reading frame of *mPlum*. mPlum.Hx Substrate Containing a Site-Specific Hypoxanthine (Hx) was generated as follows; single-stranded DNA was obtained by nicking the plasmid pmax:mPlum with Nb.BtsI (NEB R0707S) for 4 h at 37 °C in NEB Buffer 4 (NEB B7004). After phenol chloroform extraction the DNA was incubated with 5 U of ExoIII (NEB M0206S) per μg of DNA at 37 °C for 1 h in NEB buffer 1 (NEB B7001) to digest the nicked strand. 25 μg of the resulting single-strand DNA (ssDNA) were combined with 0.5 μM phosphorylated Hx-containing oligonucleotide (5′- CACG-TAGGCCTTGGXGCCGTACATGATCTG-3′, where X = Hx) in annealing buffer (1 mM Tris pH 7.5, 10 mM MgCl$_2$, 250 mM NaCl). The mixture was heated to 95 °C in a thermal cycler for 10 min, and then allowed to anneal by cooling to 27 °C. After phenol chloroform extraction, 25 μg of the annealing product was combined with 30U of T7 polymerase (NEB 0274 L) and 0.8 mM dNTPs, and incubated for 30 min at 37 °C to allow primer extension. The resulting product was then phenol chloroform extracted. In all, 10 μg of the extended nicked dsDNA was incubated with T4 DNA ligase buffer supplemented with 0.8 mM dNTPs, 0.8 mM ATP, 3U T4 DNA polymerase (NEB M0203S), and 400U T4 DNA ligase (NEB M0202S) for 10 min at 37 °C and 120 min at 16 °C for a total of three cycles to yield closed circular plasmid. Finally, the product was purified via gel extraction. In case of inefficient repair, Hx is maintained in the plasmid and due to transcriptional mutagenesis at the site of lesion results in the expression of a non-fluorescent reporter protein. Fluorescent mPlum is generated only upon efficient BER and removal of the base lesion. The mPlum fluorescence is thus proportional to the level of AAG-initiated BER. Transfected cells were analyzed for fluorescence on a BD LSR II cytometer (BD biosciences), the cell debris, doublets, and aggregates were excluded based on their side-scatter and forward-scatter properties using BD FACSDiva v6.1. To exclude dead cells Zombie NIR (BioLegend) dye was added to cells 15–30 min before the analysis. The following fluorophores and corresponding detectors (in parentheses) were used: EGFP (Alexa Fluor 488), mPlum (PE-Cy5-5), and Zombie NIR (Alexa Fluor 700). Results were computed using FlowJo$^{TM}$ v10.6.1 and Percent Fluorescent Reporter Expression %R. E. calculated after gating (Supplementary Fig. 7c–f). Fluorescent signal was $(F)$ was calculated using Eq. 1:

$$F = \frac{(N \times \text{MFI})}{S}$$

Where $N$ is the total number of live cells, MFI is the mean fluorescence intensity of the $N$ cells, and $S$ is the total number of the live cells. The fluoresce signal from the reporter EGFP expressed from the undamaged plasmid, included in all transfections as transfection efficiency control, was designated as $F^E$. The normalized fluorescence signal for EGFP reporter $F^O$ was calculated using Eq. 2:

$$F^o = \frac{F}{F^E}$$

Normalized reporter mPlum expression from the damaged reporter plamid bearing Hx $F^O_{dam}$, and that from the same reporter plasmid in the absence of damage $F^O_{un}$, were used to compute the percent reporter expression (%R.E.) using Eq. 3:

$$\%\text{R.E.} = \left(\frac{F^o_{dam}}{F^o_{un}}\right) \times 100$$

**Comet FLARE**. The level of AAG-specific substrates in actively transcribing and inhibited cells was assessed by single-cell AAG-Comet FLARE (Fragment Length Analysis using Repair Enzymes) assay. Briefly, transcriptional stalling was induced by treatment of HEK293T WT cells with 150 μM DRB for 4 h at 37 °C, DMSO served as control. Next cells were collected and mixed with 1% of low melting point agarose, spread onto glass slides (Trevigen) and lysed overnight in lysis buffer (2.5 M NaCl, 100 mM EDTA, 10 mM Tris, 1% Triton, 10% DMSO, 1% sodium lauryl sarcosinate). Next, detergent was removed by washing and the slides were incubated for 1 h at 37 °C in the presence of 2U of AAG (NEB, M0313) in reaction buffer (2.5 mM HEPES-KOH pH 7.4, 100 mM KCl, 25 mM EDTA, 2 mM BSA). The DNA was subjected to alkaline electrophoresis (1 mM EDTA, 200 mM NaOH, pH > 13) and labeled with SYBR® Green (Sigma Aldrich). In total, four replicates from two independent experiments were performed, and on average one hundred cells (fifty cells per replicate) analyzed per experiment. Slides were imaged on a Zeiss Axiovert 200 M epifluorescent microscope using a ×20 objective and the percentage of the DNA in the tail was evaluated for each data point with Comet IV, v4.2 (Perceptive Instruments, Suffolk, U.K.).

**Quantification and statistical analysis**. Immunoblots were quantified by GelEval v1.35 scientific imaging software (FrogDance Software, UK). Analysis of data was performed using GraphPad Prism v5 (GraphPad Software, Inc., La Jolla, CA). Statistical significance was determined primarily by one-way ANOVA with follow-up Dunnett's multiple comparison test (Figs. 3g–j, 4b–o, and 5i, j) and two-tailed unpaired Student's *t*-test. Comet FLARE was analyzed using one-way ANOVA and Bonferroni post hoc test. All data represent mean values ± SEM. *$p \leq 0.05$; **$p \leq 0.01$; ***$p \leq 0.001$; ****$p \leq 0.0001$ vs. control; NS, not significant.

**Reporting summary**. Further information on research design is available in the Nature Research Reporting Summary linked to this article.

## Data availability
A reporting summary is available as a Supplementary Information file. The source data underlying Figs. 1a–d, 2a–f, 3a, g–j, 4, 5, 6, and Supplementary Figs. 2a, b, g, 4, 5b, c, 6, 7a, b, and 8 are provided as a Source Data file. The RNA sequencing data reported in this paper are available in GEO under accession GSE129009 for HEK293T cells and GSE129010 for HAP1 cells. The proteomics data associated with Supplementary Data 2 are available in PRIDE under accession PXD013508. All data is available from the corresponding authors upon reasonable request.

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

## Acknowledgements

This work has been supported by the Research Council of Norway grant (ID: 263152/F20) and Swiss National Science Foundation grant (ID: 31003A_152621) to B.v.L, N.P.M. and A.B.; Onsager Fellowship to B.v.L; NTNU to D.L.B., S.B., P.A.A., L.C.O., N.K., M.O., P.S. and M.B.; University of Zurich to M.R. and A.F. We thank Raffaella Santoro, Eva Vollenweider, and Sergio Leone for help with initial ChIP experiments and RNA-sequencing analysis; to Hilde Nilsen, Lisa Lirussi, and Alexander Chaim for suggestions and shearing protocols and reagents; to Cathrine Broberg Vågbø and PROMEC for analysis of base modifications; to Hans E. Krokan, Katja Scheffler and Cathrine Broberg Vågbø, for discussions and comments to the manuscript; and to FGCZ for RNA-sequencing.

## Author contributions

B.v.L. and L.D.S. had the original idea. B.v.L. supervised the project and together with N.P.M. and M.B. interpreted the experiments. B.v.L., N.P.M., and D.L.B. wrote the manuscript and generated figures. N.P.M. performed most of the experiments D.L.B. performed gel-shift experiments, qPCR analysis of AAG substrates and Comet-FLARE. A.B. contributed to PLA, ChIP and IP experiments. M.R. purified proteins and contributed to IP studies. S.B. contributed to qPCR analysis. K.Ø.B and M.O. contributed with RNA sequencing of HAP1 cells. P.A.A. contributed to generation of HA-ELP1 cells. A.F. performed isolation of complexes. N.K. contributed to imaging. S.F.M., L.C.O., and P.S. contributed to bioinformatics analysis.

## Competing interests

The authors declare no competing interests.
