## [Peer Review File · Nature Communications]

Reviewers' comments:

Reviewer #1 (Remarks to the Author):

While the biochemical features of base excision repair have been well worked out with purified proteins and DNA substrates, the exact nature of these molecular events in chromatinized DNA in a living cell nucleus is not clear. This exciting team have found that the human alkyladenine glycosylase AAG or MPG. Interacts with a protein complex Elongator and RNAP II and helps facilitate removal of methylated bases at the 3' end of specific genes. After first showing that AAG associated with active transcription and differentially regulates a set of genes, the authors show direct interaction with Elongator and RNAP II subunits, through Co-IP. They then show using ChIP analysis that genes which are affected by the loss of AAG seem to have an 2-4 fold accumulation of AAG, ELP1, and APE1 in the 3' region of the gene. The addition of a negative control gene in these experiments is a very important addition. They then show that Elongator complex is required for the accumulation of AAG and APE, and that transcriptional repression results in decreased occupancy by AAG. Finally using two different DNA repair assays, they show that ELP1 KO or transcriptional repression affects the removal of lesions acted on by AAG. The experiments are well described and presented in a clear manner. The writing is strong and the conclusions reached are supported by these novel results. This is an important and timely study that will have a lasting impact on the field. The authors should consider the following points which would provide more impact to an already strong study:

1. The authors nicely show in the supplement that AAG interacts directly with ELP1 by Co-IP doing RNAase and Mnase and DNAase controls. It is not clear whether similar controls were performed on the AAG and RNAP polymerase shown in Figure 1.
2. Was DNAase I included prior to the LC/MS-MS experiments to rule out CO-IP through DNA?
3. What is the relative lesion frequency after this dose and treatment of MMS? Is this lesion frequency sufficient to hit the gene targets in this study
4. Do the authors have any evidence that the unstructured 80 AA of AAG are sufficient to bring down ELP1? For example, would these 80 AA fused to GFP be necessary and sufficient to bring down ELP1?
5. Have the authors tried to use proximity ligase assay to show direct association of these proteins ?
6. The COMET-FLARE assay shown in 5e, looks like only one time point.
7. Inclusion of a summary figure showing a working model based on the results of this study would be very helpful as this study once published will have an important impact on the field.

Some more minor concerns, include:

1. While AAG repairs methylated bases it also removes ethenoA and hypoxanthine, bases that can arise from oxidized lipids or nitric oxide, respectively – this should be included in the introduction. This seems especially irrelevant with respect to the FM-HCR assay shown in Figure 4. Is it possible that these lesions might also be responsible for the 3' enrichment at certain genes?
2. In the introduction the authors state that, "This idea is further supported by the notion that BER preferentially occurs on the transcribed strand." The author should include the elegant work by Spivak using COMET-FISH to show strand specific repair of 8-oxoG. Guo J, Hanawalt PC, Spivak G. Comet-FISH with strand-specific probes reveals transcription-coupled repair of 8-oxoGuanine in human cells. *Nucleic Acids Res.* 2013 Sep; 41(16):7700-12. doi: 10.1093/nar/gkt524. Epub 2013 Jun 17. PubMed PMID: 23775797; PubMed Central PMCID: PMC3763531.
3. Please give the number of significant genes and size of each class when showing enriched gene ontology terms in Figures 1 and 2. While the P values are clearly impressive if the overall number of genes in the class is low, then one wonders about the biological significance.
4. Since the G in AAG stands for glycosylase, saying AAG DNA glycosylase is a bit redundant. Is this typical in the literature?

Reviewer #2 (Remarks to the Author):

In this manuscript, the authors investigated the functional relationship between AAG and a complex with actively transcribing RNA pol II. They clearly showed the direct interaction of AAG with the ELP1 subunit of transcriptional Elongator complex by using purified recombinant proteins. Furthermore, they generated AAG^{-/-}, ELP1^{-/-}, and AAG^{-/-}ELP1^{-/-} cells and then found AAG and ELP1 co-regulated genes whose 3'ends are occupied with AAG, ELP1 and RNA pol II through the combined analysis of RNA-Seq and ChIP-Seq. In addition, they showed that the AAG occupancy depends on ELP1 and AAG-initiated BER is coupled with active transcription. The experiments are well executed and logically analyzed, which makes an important contribution to understanding the mechanisms of the AAG-initiated BER. However, further molecular insight and the role of the direct interaction between AAG and ELP1 are required for publication with a novel concept.

Major points:

1. Although the authors showed that the function of AAG depends on ELP1, using the ELP1^{-/-} cells, there is a lack of evidence about the necessity of their direct interaction. Since ELP1 is essential for the assembly and activity of the Elongator complex, there remains the possibility that AAG is associated with Elongator complex through the other component, not ELP1. Therefore, the authors should determine the region of AAG responsible for the binding to ELP1, and they should use the ELP1-binding-deficient AAG to evaluate the necessity of its ELP1 binding for AAG-initiated BER. At least the authors should test whether the phenotypes of ELP1^{-/-} cells are rescued by re-expression of wild-type ELP1 and $\Delta 80$ ELP1 mutant.
2. Through the AAG ChIP-Seq analysis in WT and ELP1^{-/-} cells, the authors showed that loss of ELP1 causes global reduction in AAG binding to the chromatin at all genes tested in this study, and therefore proposed that ELP1 is essential for AAG chromatin recruitment. If it is true, the subcellular localization of AAG is expected to change in ELP1^{-/-} cells. Thus, the authors should test it by immunofluorescence microscopy or chromatin fractionation assay of WT and ELP1^{-/-} cells.
3. In the manuscript, the authors described that AAG and APE1 distribution patterns at the 3'end of the co-regulated genes are similar to ELP1 and RNA pol II distribution patterns. However, there is a lack of evidence that their ChIP-Seq peaks are overlapped with each other at the chromosome regions. Therefore, the authors should clearly indicate whether AAG1 are colocalized with ELP1, APE1, and RNA pol II at the 3'end of the co-regulated genes through their ChIP-Seq profiles.

Minor points:

1. In this manuscript, the reason that $\Delta 80$ ELP1 mutant was used is not described.
2. Fig. 2k-n, 3, 4g, 5e and Supplementary fig. 3 and 4: The authors can use other efficient statistical analyses of multiple data. What is **** ?
3. Fig. 2f and i: Labels are missing, compared with Fig. 1f and h.
4. Fig.3: It is better to show the sites tested in gene structure.

Reviewer #3 (Remarks to the Author):

"Alkyladenine DNA glycosylase associates with transcription elongation to coordinate DNA repair with gene expression" by Montaldo et al.

This paper examines the coordination between DNA transcription and repair and investigates epigenetic mechanisms involved in this process. Immunoprecipitation experiments provide evidence for interactions of AAG repair protein with hyperphosphorylated RNA pol II.

RNAseq experiments revealed that many genes were differentially expressed in AAG^{-/-} cells, many of the affected genes are involved in neurogenesis. The authors interpret this to indicate that AAG

forms a complex with active transcription machinery. However, changes in gene expression were relatively mild, and this result can be explained by many difference mechanisms other than direct interactions of AAG with the transcription machinery. For example, accumulation of DNA lesions in repair deficient cells may interfere with gene expression or signal for up regulation of additional DNA repair genes. Therefore, the conclusions are not fully supported by the data shown.

LC-MS experiments were used to identify proteins that form complex with AAG, but the experiments are not explained well. Since AAG was FLAG tagged, I assume they conducted some sort of affinity pulldown: this is not mentioned in the main text or in the supplement. What was used as negative control? How were protein amounts quantified? These details are not given so it is difficult to evaluate the data in Figure 2. Furthermore, affinity pulldown cannot identify direct binders because they cannot be distinguished from secondary interacting proteins. A crosslinking study would be more convincing. It is not clear how silver stained proteins (Fig. 2A) could have been used in proteomics experiments as silver staining is not compatible with mass spectrometry analyses. It is somewhat surprising only 4 proteins were identified in this experiment (Fig. 2B). MMS concentrations used (2 mM) are exceedingly high and likely kill most of the cells. Interestingly, AAG lacking 80 N-terminal amino acids still binds ELP1, and other regions were not investigated (Fig. 2d)

RNA seq experiments shown in Fig 2 show an opposite trend for genes co-regulated genes in AAG^{-/-} and EPL^{-/-} cells - this result is puzzling as it is proposed that the interaction between the two proteins is important for the same biological process.

Chip Seq results suggest that AAG and HA-ELP1 occupancy is increased at 3' ends of coregulated genes, which coincided with locations of AAG substrates. However the authors did not compare the Chip Seq results with their RNA-seq data, e.g is there a correlation between occupancy and gene expression?

The authors use a term "aberrant bases" for AAG substrates, this term is too vague as there are hundreds of aberrant bases known.

Overall, this reviewer is not convinced that BER and transcription are interconnected as claimed. While this hypothesis is interesting, the data presented are somewhat indirect/open ended and do not tell the full story. The initial results for binding should be confirmed by gel shift experiments with recombinant proteins (ELP1 and AAG) and chemical crosslinking.

The study lacks any sort of structural perspective. Why did they choose to delete 80 N-terminal amino acids of AAG? What domain do this correspond to? No structural models are provided. In regard to writing style, the manuscript is not written clearly. There are many sentences that are too long or are awkwardly constructed, and many of the experiments are not explained well.

Overall, I do not think that this manuscript is of high enough caliber to be published in Nature Communications.

Detailed point by point response to reviewers' comments:

Reviewer #1

Answers to reviewer #1 comments:

Point 1. The authors nicely show in the supplement that AAG interacts directly with ELP1 by Co-IP doing RNAase and Mnase and DNAase controls. It is not clear whether similar controls were performed on the AAG and RNAP polymerase shown in Figure 1.

Answer: We thank reviewer for the suggested experiment. We have now performed AAG Co-IP experiments in the presence of RNase, Mnase and DNaseI (Fig. 1c). The results indicate that presence of nucleases does not affect the amount of active RNA polymerase II in complex with AAG. This is in line with the observation that AAG interacts with ELP1 independently of the presence of nucleic acids (Supplementary figure 2b,c). Taken together, these results suggest that AAG and RNA polymerase II complex formation is a result of protein-protein interactions independent of the presence of nucleic acids. Description of the results has been included in the revised manuscript.

Point 2. Was DNAase I included prior to the LC/MS-MS experiments to rule out CO-IP through DNA?

Answer: DNaseI was not added prior to LC/MS-MS, since the main aim was to determine general cellular complexes that contain FLAG-AAG, and might be relevant both in the absence and presence of DNA. That AAG and ELP1 interact in the absence of DNA we demonstrated by Co-IP experiments in the presence of RNase, Mnase and DNaseI (Supplementary figure 2b,c). Further, we have validated that these two proteins (ELP1 and AAG) directly interact, by using purified recombinant proteins in Co-IP's (Fig. 2f), and newly included gel shift experiments combined with chemical crosslinking (Fig. 2d). See for more details response to reviewer 3.

Point 3. What is the relative lesion frequency after this dose and treatment of MMS? Is this lesion frequency sufficient to hit the gene targets in this study?

Answer: The experiments in this manuscript have almost all been performed under conditions without MMS treatment. Only experiments done in the presence of alkylation exposure were the proteomic study and the Co-IP presented in Fig. 2a-c. Our findings indicate that under normal growth conditions without MMS treatment relative amount of aberrant bases, substrates of AAG, is increased towards the 3'-end of gene targets (Fig. 4m-p). Currently however, the methods to determine the frequency of aberrant bases, which are removed by AAG, on the level of specific genes both under exposure or physiological conditions are lacking. Due to observed uneven distribution of aberrant bases throughout the genome (Fig. 4, and previous findings: Mao, et.al. Genome Res. 2017. 27: 1674–1684; Ding, et.al. J Am Chem Soc. 2017. 139: 2569–2572; Wu, et.al. J Am Chem Soc. 2018. 140: 9783–9787; Poetsch, et.al. Genome Biol. 2018. 19: 215), it is unfortunately not possible to estimate the frequency based on the measurements of the global levels in genomic DNA.

Point 4. Do the authors have any evidence that the unstructured 80 AA of AAG are sufficient to bring down ELP1? For example, would these 80 AA fused to GFP be necessary and sufficient to bring down ELP1?

Answer: To address reviewers helpful suggestion we complemented HEK293T AAG^{-/-} cells with either GFP-1-80aa AAG or GFP alone (Fig. 2g), and performed subsequent Co-IPs. Indeed,

presence of 80 N-terminal amino acids (aa) of AAG was sufficient to bring down ELP1. As expected no ELP1 was detected in Co-IP using cells expressing GFP alone. Similar result was observed when GFP was fused to the C-term of 1-80aa AAG (Supplementary figure 2g), thus further indicating that the position of GFP tag does not influence the interaction. Taken together, results of this important experiment indicate that 80 N-terminal amino acids of AAG is sufficient to co-IP ELP1, and as such support the conclusion that the unstructured region of AAG is engaged in the interaction with ELP1.

Point 5. Have the authors tried to use proximity ligase assay to show direct association of these proteins?

Answer: Following suggestion from the reviewer we performed proximity ligation assays (PLA) targeting ELP1 subunit of Elongator complex and AAG. Using PLA approach we were able to successfully visualize association between AAG and ELP1 (Fig. 2h) that occurs primarily in the nucleus. Negative controls are included in Supplementary figure 2h,i. Taken together, findings of PLA experiments provide additional evidence that AAG and ELP1 directly interact. The results have been included in the revised manuscript.

Point 6. The COMET-FLARE assay shown in 5e, looks like only one time point.

Answer: The impact of transcription elongation inhibition on the repair of aberrant bases, substrates of AAG, was analyzed 4 hours after the initiation of DRB treatment. This time point is in accordance with previous studies that successfully applied DRB treatment to inhibit transcription elongation (some of the published works include, Amir-Zilberstein, et.al. Mol Cell Biol. 2007. 27: 5246–5259.; Pawellek, et.al. JBC 2014. 289: 34683–34698; Day, et.al. Methods 2016. 96: 59–68.; Baluapuri, et.al. Mol.Cell 2019. 74: 674-687; etc.).

Point 7. Inclusion of a summary figure showing a working model based on the results of this study would be very helpful as this study once published will have an important impact on the field.

Answer: We are grateful for the reviewers remark. A working model has been included as summary figure 7 and the details described in the revised manuscript.

Minor points:

1. While AAG repairs methylated bases it also removes ethenoA and hypoxanthine, bases that can arise from oxidized lipids or nitric oxide, respectively – this should be included in the introduction. This seems especially irrelevant with respect to the FM-HCR assay shown in Figure 4. Is it possible that these lesions might also be responsible for the 3' enrichment at certain genes?

Answer: The information about spectra of aberrant bases recognized by AAG has been included in the introduction.

While we cannot exclude a possibility that ethenoA and hypoxanthine contribute the AAG 3' enrichment at the specific genes, Lindahl and Barnes D.E. suggested that nonenzymatic methylation by S-adenosylmethionine results in ~7200 aberrantly methylated bases (primarily 7meG and 3meA) per mammalian cell in 24h; in contrast levels of ethenoA and hypoxanthine were estimated to be approximately 5-fold lower (combined ~1500 per cell per day) (Lindahl and Barnes. Cold Spring Harbor symposia on quantitative biology. 2000. 65: 127–133; Friedberg et.al. 2006. "DNA repair and mutagenesis" ASM Press.). The indicated higher incidence of aberrantly

methylated bases, when compared to other AAG substrates, is a basis for suggestion that this type of base lesions is most likely responsible for the 3' enrichment.

2. In the introduction the authors state that, “This idea is further supported by the notion that BER preferentially occurs on the transcribed strand.” The author should include the elegant work by Spivak using COMET-FISH to show strand specific repair of 8-oxoG. Guo J, Hanawalt PC, Spivak G. Comet-FISH with strand-specific probes reveals transcription-coupled repair of 8-oxoGuanine in human cells. Nucleic Acids Res. 2013 Sep;41(16):7700-12. doi: 10.1093/nar/gkt524. Epub 2013 Jun 17. PubMed PMID: 23775797; PubMed Central PMCID: PMC3763531.

Answer: We thank the reviewer for pointing at this important work, which we have now included in the manuscript.

3. Please give the number of significant genes and size of each class when showing enriched gene ontology terms in Figures 1 and 2. While the P values are clearly impressive if the overall number of genes in the class is low, then one wonders about the biological significance.

Answer: Detailed number of significant DEGs and the size of each class indicated in enriched gene ontology terms (depicted in revised Figures 1 and 3) is presented in Supplementary table 1. On average DEGs segregating to top enriched gene ontology terms compose ~20% of all DEGs.

4. Since the G in AAG stands for glycosylase, saying AAG DNA glycosylase is a bit redundant. Is this typical in the literature?

Answer: The change has been implemented.

Reviewer #2:

Answers to reviewer #2 comments:

Point 1. Although the authors showed that the function of AAG depends on ELP1, using the ELP1^{-/-} cells, there is a lack of evidence about the necessity of their direct interaction. Since ELP1 is essential for the assembly and activity of the Elongator complex, there remains the possibility that AAG is associated with Elongator complex through the other component, not ELP1. Therefore, the authors should determine the region of AAG responsible for the binding to ELP1,

Answer: The initial pull-down experiments suggested that in the absence of unstructured N-terminal region (1-80aa), catalytic domain of AAG is unable to efficiently interact with ELP1 (Fig. 2f). Following helpful suggestion of the reviewer to further determine the region of AAG responsible for interaction with ELP1, we complemented HEK293T AAG^{-/-} cells with 1-80aa AAG peptide fused with GFP (Fig. 2g and Supplementary figure 2g) and performed Co-IP experiments. In line with the initial pull-down results, the unstructured N-terminal region of AAG was sufficient to IP ELP1, thus indicating that the N-terminal region of AAG is responsible for interaction with ELP1. Taken together these results provide additional molecular insight into the interaction between these two proteins. See also answer to the point 4 of reviewer 1.

and they should use the ELP1-binding-deficient AAG to evaluate the necessity of its ELP1 binding for AAG-initiated BER. At least the authors should test whether the phenotypes of ELP1^{-/-} cells are rescued by re-expression of wild-type ELP1 and Δ80 ELP1 mutant.

Answer: To address this important remark we examined the impact of ELP1 binding for AAG-initiated BER by determining BER capacity in HEK293T AAG^{-/-} cells complemented with full-length GFP-AAG or GFP-Δ80AAG lacking region important for interaction with ELP1 (Fig. 5j). Both proteins were GFP-tagged, allowing to directly relate BER capacity with the amount of protein present in the cells. Further, both full-length GFP-AAG and GFP-Δ80AAG were expressed at the level comparable to endogenous AAG present in the HEK293T WT cells (Supplementary figure 7b). Previous work indicated that both AAG forms have comparable catalytic activity (O'Connor et.al. 1993; O'Brien et. Al 2003). Interestingly, while expression of full-length AAG successfully rescued BER capacity, the AAG-initiated BER remained significantly reduced in HEK293T AAG^{-/-} cells expressing Δ80AAG, when compared to HEK293T WT cells. Taken together, these findings provide insight into the functional importance of the interaction between AAG and ELP1.

Point 2. Through the AAG ChIP-Seq analysis in WT and ELP1^{-/-} cells, the authors showed that loss of ELP1 causes global reduction in AAG binding to the chromatin at all genes tested in this study, and therefore proposed that ELP1 is essential for AAG chromatin recruitment. If it is true, the subcellular localization of AAG is expected to change in ELP1^{-/-} cells. Thus, the authors should test it by immunofluorescence microscopy or chromatin fractionation assay of WT and ELP1^{-/-} cells.

Answer: To address the reviewers suggestion we compared AAG presence in chromatin bound fraction of HEK293T WT, and ELP1^{-/-} cells. In line with ChIP-qPCR results (Fig. 5 a-d) we observed significant reduction in AAG levels in ELP1^{-/-}, when compared to WT chromatin fraction (Fig. 5e and f). This effect was chromatin specific, since no difference in global AAG levels was observed between the two genotypes. Taken together, these experiments thus further support the idea that ELP1 plays an important role in AAG recruitment to the chromatin.

Point 3. In the manuscript, the authors described that AAG and APE1 distribution patterns at the 3'end of the co-regulated genes are similar to ELP1 and RNA pol II distribution patterns. However, there is a lack of evidence that their ChIP-Seq peaks are overlapped with each other at the chromosome regions. Therefore, the authors should clearly indicate whether AAG1 are colocalized with ELP1, APE1, and RNA pol II at the 3'end of the co-regulated genes through their ChIP-Seq profiles.

Answer: As reviewer insightfully pointed it is very important to ensure that all analyzed factors colocalize in precisely defined gene regions. In Fig. 4 (former Fig. 3) the distribution of AAG, APE1, ELP1 and RNA pol II was investigated using ChIP-qPCR analysis. This approach allowed us to test the distribution of BER and transcription components in tightly defined gene regions. The distribution of AAG, APE1, ELP1 and RNA pol II, as well as of the aberrant bases, was analyzed in 3 out of 41 co-regulated genes with the same expression directionality, thus nearly 10% (Fig. 3f). All of the three genes showed enrichment of AAG, ELP1, APE1 and aberrant bases in the assayed region of the genes' 3' end compared to the genes' 5' end. As a control, we investigated colocalization and accumulation in a gene that was not co-regulated (YTHDC1); this gene did not show the defined 3'-end enrichment as shown for the co-regulated genes (Fig. 4 e-m). We have for clarity revised the text, placing an emphasis on the defined regions, as well as have included the schematic representation of the tested gene regions.

Minor points:

1. In this manuscript, the reason that $\Delta 80$ ELP1 mutant was used is not described.

Answer: We presume that the reviewer suggests to elaborate on the use of $\Delta 80$ AAG. The manuscript has been revised in accordance with the suggestion. Briefly, since AAG is composed of the unstructured N-terminal region and the catalytic DNA glycosylase domain, we generated the $\Delta 80$ AAG to test if the catalytic domain per se is sufficient for the interaction with ELP1.

2. Fig. 2k-n, 3, 4g, 5e and Supplementary fig. 3 and 4: The authors can use other efficient statistical analyses of multiple data. What is ** ?**

Answer: Following reviewers suggestion we have reanalyzed suggested results by applying one-way ANOVA. p-values have been updated including **** $p \leq 0.0001$.

3. Fig. 2f and i: Labels are missing, compared with Fig. 1f and h.

Answer: Labels have been added in the revised Fig. 3b and e.

4. Fig.3: It is better to show the sites tested in gene structure.

Answer: We thank reviewer for the helpful suggestion. The site representation has been included in the updated Fig. 4.

Reviewer #3 (Remarks to the Author):

Answers to reviewer #3 comments:

“Alkyladenine DNA glycosylase associates with transcription elongation to coordinate DNA repair with gene expression” by Montaldo et al. This paper examines the coordination between DNA transcription and repair and investigates epigenetic mechanisms involved in this process. Immunoprecipitation experiments provide evidence for interactions of AAG repair protein with hyperphosphorylated RNA pol II. RNAseq experiments revealed that many genes were differentially expressed in AAG^{-/-} cells, many of the affected genes are involved in neurogenesis. The authors interpret this to indicate that AAG forms a complex with active transcription machinery. However, changes in gene expression were relatively mild, and this result can be explained by many difference mechanisms other than direct interactions of AAAG with the transcription machinery. For example, accumulation of DNA lesions in repair deficient cells may interfere with gene expression or signal for up regulation of additional DNA repair genes. Therefore, the conclusions are not fully supported by the data shown.

Answer: We thank the reviewer for the comment. As depicted in Fig. 3g-j and Supplementary figure 3, the changes in the expression of specific neurodevelopmental genes observed in AAG^{-/-} cells range from 1.5- to 5- fold. The size of these changes in the co-regulated genes is thus very similar to the changes determined in cells lacking ELP1, the subunit of transcriptional Elongator complex with importance in neurodevelopment (Fig. 3). Since HEK293T cells only may have certain neural features, and are not neuronal cell model, they do not allow addressing the full impact of transcription associated AAG-initiated BER on the expression of neurodevelopmental genes and processes. We agree with the reviewer that the exciting next step is to explore the role of AAG-initiated BER in neurodevelopment (as indicated in the discussion “It will be thus interesting to determine the importance of transcription associated AAG-initiated BER in regulation

of neurodevelopment, and test its role in brain functioning.”), a question that merits investigation in future work.

In addition, following reviewers suggestion, we tested if the expression of additional DNA repair genes is altered (2-fold or higher) in AAG^{-/-} cells. Analysis of RNAseq results did not reveal any significant change in the expression of DNA repair genes (all results are available in GEO under accession GSE129009).

LC-MS experiments were used to identify proteins that form complex with AAG, but the experiments are not explained well. Since AAG was FLAG tagged, I assume they conducted some sort of affinity pulldown: this is not mentioned in the main text or in the supplement. What was used as negative control? How were protein amounts quantified? These details are not given so it is difficult to evaluate the data in Figure 2. Furthermore, affinity pulldown cannot identify direct binders because they cannot be distinguished from secondary interacting proteins. A crosslinking study would be more convincing. It is not clear how silver stained proteins (Fig. 2A) could have been used in proteomics experiments as silver staining is not compatible with mass spectrometry analyses. It is somewhat surprising only 4 proteins were identified in this experiment (Fig. 2B). MMS concentrations used (2 mM) are exceedingly high and likely kill most of the cells. Interestingly, AAG lacking 80 N-terminal amino acids still binds ELP1, and other regions were not investigated (Fig. 2d)

Answer: We apologize for lack of clarity. To identify proteins that form a complex with AAG, we expressed and affinity-purified the FLAG-tagged AAG from HEK293T cells, either untreated or exposed to the alkylating agent MMS. A control mock purification was performed in parallel from cells transfected with empty FLAG vector. The affinity-purified samples were next subjected to liquid chromatography-tandem mass spectrometry (LC/MS-MS) analysis (Fig. 2b, Supplementary table 2). All MS/MS results were analyzed using Sequest; detailed description is included in revised material and methods. Using this approach ELP1 was identified as the most enriched novel interacting partner in AAG containing samples. In the same samples ELP 2 and 3 subunits have been detected, suggesting presence of the whole holo-Elongator. All remaining proteins identified by LC/MS-MS are listed in Supplementary table 2, and the results deposited in PRIDE under accession PXD013508.

Silver staining analysis was performed solely to visualize proteins in different samples (Fig. 2a).

MMS has been applied for 1h, at dose that has been reported previously (Zhang, et.al. JBC. 2007. 282: 15330–15340). While MMS exposure was not central to this study, the results presented in Fig. 2b,c indicate that even increased MMS doses, do not exacerbate the interaction.

To address reviewers concern and to exclude the possibility that binding of AAG to ELP1 is a consequence of secondary interactions we have performed gel-shift experiments in the presence of cross-linking agent (Fig. 2d). Please see detailed response below.

In addition to confirm that the 80 N-terminal amino acids of AAG are crucial for the interaction with ELP1, we expressed 1-80aa AAG GFP-tagged in HEK293T AAG^{-/-} cells (Fig. 2g and Supplementary figure 2g) and performed Co-IP experiments. Importantly, the unstructured N-terminal region of AAG was sufficient to IP ELP1, indicating that this region of AAG mediates the interaction with ELP1. See also answer to the point 4 of reviewer 1, and point 1 of reviewer 2.

RNA seq experiments shown in Fig 2 show an opposite trend for genes co-regulated genes in AAG^{-/-} and EPL^{-/-} cells - this result is puzzling as it is proposed that the interaction between the two proteins is important for the same biological process.

Answer: We thank reviewer for this comment. Our findings indeed suggest that AAG through its unstructured N-terminal region associates with the ELP1 subunit of Elongator and forms complex with the active transcription machinery (Fig. 1 and 2). As a consequence of active transcription chromatin is suggested to be locally decondensed, which allows AAG to efficiently initiate BER by recognizing and removing aberrant bases. BER initiation likely temporarily inhibits RNA pol II progression, resulting in reduced expression of co-regulated genes (Fig. 3 and Supplementary figures 3,4 and 6). In the absence of Elongator, transcription of target genes is repressed due to reduced elongation (Close et. al. Mol.Cell. 2006. 22: 521-531), while AAG chromatin recruitment and initiation of BER are impaired (Fig. 5). Taken together, our results suggest that Elongator recruits AAG during transcription, and is in line with the finding that gene expression is repressed in AAG^{-/-}/ELP1^{-/-} cells, similar to ELP1^{-/-} cells (Fig. 3). For clarity, and following suggestion of Reviewer 1, we have included a model in Fig. 7.

Chip Seq results suggest that AAG and HA-ELP1 occupancy is increased at 3' ends of coregulated genes, which coincided with locations of AAG substrates. However the authors did not compare the Chip Seq results with their RNA-seq data, e.g is there a correlation between occupancy and gene expression?

Answer: The results of ChIP-qPCR analysis (Fig. 4) indicated that there is a direct correlation between AAG, HA-ELP1 occupancy and the expression of co-regulated genes (Fig. 3). AAG and HA-ELP1, as well as APE1 and active RNA pol II (Fig.4 and Supplementary figure 6), all clearly accumulated in defined regions towards 3'end of nearly 10% tested co-regulated genes (3 out of 41), which presented with the same expression directionality. Based on the observed clear and distinct distribution pattern of AAG, APE1 and ELP1 at co-regulated genes, we believe that current results provide sufficient answers to this study. While we feel that the ChIPseq experiments are beyond the scope of this study, we aim to perform this analysis in a follow up work.

The authors use a term "aberrant bases" for AAG substrates, this term is too vague as there are hundreds of aberrant bases known.

Answer: We apologize for possible lack of clarity. We decided to use the term aberrantly methylated bases, since AAG is the only DNA glycosylase known to act on the two most predominant methylated bases, 7meG and 3meA.

Overall, this reviewer is not convinced that BER and transcription are interconnected as claimed. While this hypothesis is interesting, the data presented are somewhat indirect/open ended and do not tell the full story. The initial results for binding should be confirmed by gel shift experiments with recombinant proteins (ELP1 and AAG) and chemical crosslinking.

Answer: Following reviewers insightful suggestion we have performed gel shift experiments using purified AAG and ELP1, and chemical crosslinking with bisulfosuccinimidyl suberate (BS³). Importantly, the gel shift experiments combined with chemical crosslinking revealed that AAG efficiently forms a complex with ELP1 (Fig. 2d, lanes 4,5). In agreement with previous work, ELP1 efficiently dimerized (Xu, PNAS 2015 10697; Dauden EMBO 2017, 264), a property important for Elongator assembly. No complex formation was observed in control reactions with crosslinker and AAG alone (Fig. 2d, lane 6). Taken together, these results provide important evidence that AAG and ELP1 binding is direct and not a result of secondary interactions.

The study lacks any sort of structural perspective. Why did they choose to delete 80 N-

terminal amino acids of AAG? What domain do this correspond to? No structural models are provided.

Answer: Since AAG is composed of the unstructured N-terminal region and the catalytic DNA glycosylase domain, we decided to generate $\Delta 80$ AAG to address whether the catalytic domain is sufficient to interact with ELP1, or if the unstructured region binds to ELP1. New results presented in Fig. 2g and Supplementary Fig. 2g indicate that the N-terminal AAG region is engaged in the interaction with ELP1. Due to lack of any structural data about the 80aa N-terminal AAG and the unstructured nature of this region, we were unable to perform structural modeling.

In regard to writing style, the manuscript is not written clearly. There are many sentences that are too long or are awkwardly constructed, and many of the experiments are not explained well.

Overall, I do not think that this manuscript is of high enough caliber to be published in Nature Communications.

Answer: We thank to the reviewer for knowledgeable comments and suggested experiments. We hope that by incorporating requested changes and performing additional experiments the manuscript is significantly improved and meets the raised points.

REVIEWERS' COMMENTS:

Reviewer #1 (Remarks to the Author):

The authors have done an excellent job in responding to the reviewers concerns. The study has been strengthened by a series of new experiments and clarifications in the manuscript. This is an important and timely study. In the rebuttal to reviewer 1's concern regarding the lesion frequency, the authors suggest there are not currently any techniques to look at this at a gene level. This is not entirely true as one of the authors published with Dave Scicchitano to actually look at whether alkylation damage is repaired in a transcriptional coupled manner, in light of these new results this paper should be cited and discussed. Plosky B, Samson L, Engelward BP, Gold B, Schlaen B, Millas T, Magnotti M, Schor J, DA. Base excision repair and nucleotide excision repair contribute to the removal of N-methylpurines from active genes. DNA Repair (Amst). 2002 Aug 6; 1(8):683-96. PMID: 12509290

Reviewer #2 (Remarks to the Author):

The authors sincerely responded to reviewers' comments as possible as they could. However, scientific impact and general interest of the manuscript would be insufficient for publication in Nature Communications, because the finding of the association between BER and transcription lacks novelty and requires strong advancement in this field. Unfortunately, I think that this manuscript would be more suitable for publication in specialized journal of basic science.

Reviewer #3 (Remarks to the Author):

This article investigates a possible association of base excision repair and transcription. This work is exciting and novel and has a potential to make an important contribution to the field of DNA repair. In this revised version of the manuscript, the authors added new gel shift experiments that support their conclusions regarding interactions between AAG and RNA pol II. Overall, the revised version is stronger and the story is more complete. They have added a useful scheme and improved clarity. Although some of the issues have not been addressed, I believe there is sufficient data to support publication.

Detailed point by point response addressing a specific suggestion brought by the reviewer:

Reviewer #1

The authors have done an excellent job in responding to the reviewers concerns. The study has been strengthened by a series of new experiments and clarifications in the manuscript. This is an important and timely study.

Point 1. *In the rebuttal to reviewer 1's concern regarding the lesion frequency, the authors suggest there are not currently any techniques to look at this at a gene level. This is not entirely true as one of the authors published with Dave Scicchitano to actually look at whether alkylation damage is repaired in a transcriptional coupled manner, in light of these new results this paper should be cited and discussed. Plosky B, Samson L, Engelward BP, Gold B, Schlaen B, Millas T, Magnotti M, Schor J, DA. Base excision repair and nucleotide excision repair contribute to the removal of N-methylpurines from active genes. DNA Repair (Amst). 2002 Aug 6;1(8):683-96. PMID: 12509290*

Answers to reviewer #1 comment: We thank to the reviewer for the remark and pointing at this very important work. We have now included and discussed the study in the manuscript.

We thank to the Reviewer #2 and #3 for considering our revised manuscript. No specific points have been raised by the two reviewers.